# Convergent evolution of SARS-CoV-2 Omicron subvariants leading to the emergence of BQ.1.1 variant

Jumpei Ito [1,40], Rigel Suzuki[2,40], Keiya Uriu[1,3,40], Yukari Itakura [4,40], Jiri Zahradnik [5,6,40], Kanako Terakado Kimura[7,40], Sayaka Deguchi[8,40], Lei Wang [9,10,40], Spyros Lytras [11,40], Tomokazu Tamura[2], Izumi Kida [12], Hesham Nasser [13,14], Maya Shofa[15,16], Mst Monira Begum[13], Masumi Tsuda [9,10], Yoshitaka Oda[9], Tateki Suzuki [7], Jiei Sasaki[7], Kaori Sasaki-Tabata[17], Shigeru Fujita [1,3], Kumiko Yoshimatsu [18], Hayato Ito[2], Naganori Nao[19,25,26], Hiroyuki Asakura[20], Mami Nagashima[20], Kenji Sadamasu[20], Kazuhisa Yoshimura[20], Yuki Yamamoto[21], Tetsuharu Nagamoto[21], Jin Kuramochi [22,23], Gideon Schreiber [5], The Genotype to Phenotype Japan (G2P-Japan) Consortium*, Akatsuki Saito [15,16,24], Keita Matsuno [12,26,27], Kazuo Takayama[8,28], Takao Hashiguchi [7] ✉, Shinya Tanaka [9,10] ✉, Takasuke Fukuhara [2,28,29] ✉, Terumasa Ikeda [13] ✉ & Kei Sato [1,3,30,31,32,33,34] ✉

In late 2022, various Omicron subvariants emerged and cocirculated worldwide. These variants convergently acquired amino acid substitutions at critical residues in the spike protein, including residues R346, K444, L452, N460, and F486. Here, we characterize the convergent evolution of Omicron subvariants and the properties of one recent lineage of concern, BQ.1.1. Our phylogenetic analysis suggests that these five substitutions are recurrently acquired, particularly in younger Omicron lineages. Epidemic dynamics modelling suggests that the five substitutions increase viral fitness, and a large proportion of the fitness variation within Omicron lineages can be explained by these substitutions. Compared to BA.5, BQ.1.1 evades breakthrough BA.2 and BA.5 infection sera more efficiently, as demonstrated by neutralization assays. The pathogenicity of BQ.1.1 in hamsters is lower than that of BA.5. Our multiscale investigations illuminate the evolutionary rules governing the convergent evolution for known Omicron lineages as of 2022.

As of November 2022, the SARS-CoV-2 Omicron variant (B.1.1.529 and BA lineages) is the only current variant of concern (VOC)[1]. At the end of November 2021, Omicron BA.1 rapidly outcompeted Delta, a prior VOC. Soon after the global spread of the Omicron BA.1 lineage, Omicron BA.2 became predominant worldwide. Thereafter, a variety of

BA.2 descendants, such as BA.5 and BA.2.75, emerged and are becoming predominant in certain regions.

Since its emergence, Omicron BA.2 has highly diversified and transformed into other Omicron subvariants, including variants that have recently emerged. Although both BA.5 and BA.2.75 diversified

---

A full list of affiliations appears at the end of the paper. *A list of authors and their affiliations appears at the end of the paper.
✉e-mail: hashiguchi.takao.1a@kyoto-u.ac.jp; tanaka@med.hokudai.ac.jp; fukut@pop.med.hokudai.ac.jp; ikedat@kumamoto-u.ac.jp; KeiSato@g.ecc.u-tokyo.ac.jp

from BA.2, these two Omicron subvariants are phylogenetically independent of each other, suggesting that BA.5 and BA.2.75 emerged independently[2]. However, recent studies, including ours, have demonstrated that the spike (S) proteins of these two variants exhibit similar evolutionary patterns; one pattern is an amino acid substitution that evades humoral immunity, while the other is a substitution that increases the binding affinity to human angiotensin-converting enzyme 2 (ACE2), the receptor for SARS-CoV-2 infection. For BA.5, the F486V substitution contributes to evasion from humoral immunity[3–6], while the L452R substitution increases ACE2 binding affinity[4,7–9]. For BA.2.75, the G446S substitution is responsible for evading humoral immunity[6,10–14], while the N460K substitution increases ACE2 binding affinity[2,7,11].

Before the emergence of BA.5, newly emerging SARS-CoV-2 variants frequently outcompeted previously dominant variants every few months. However, as of November 2022, although a variety of Omicron subvariants (including BA.2.75) emerged after BA.5, none have successfully outcompeted all other variants to become the predominant lineage worldwide. Instead, recently emerged Omicron subvariants show similar fitness advantages and are cocirculating with each other. Notably, most of these cocirculating variants have convergently acquired substitutions at specific sites in the S protein,

namely, 346, 444, 452, 460, and 486. While previous studies have documented this convergent evolution of the S protein in Omicron lineages[15,16], it remains unclear (i) how frequently these convergent substitutions occur during Omicron evolution and (ii) how much these convergent substitutions contribute to the selective advantage of the circulating viruses.

BQ.1.1 is a descendant of BA.5 and bears all five recent convergent substitutions (R346T, K444T, L452R, N460K, or F486V). As of October 12, 2022, the WHO classified BQ.1.1 as an Omicron subvariant to be monitored[1]. In this study, we illuminate the evolutionary principles underlying the current convergent evolution of Omicron lineages, providing a comprehensive phylogenetic analysis, modeling analysis to predict the fitness landscape of the Omicron S protein. Furthermore, we characterized the virological properties of BQ.1.1, including its immunogenicity, fusogenicity, and pathogenicity in a hamster model without a history of vaccination and viral infection (hereafter referred to as intrinsic pathogenicity).

## Results
### Convergent evolution of Omicron lineages
As of November 2022, various Omicron lineages have continuously emerged, such as Omicron BA.1, BA.2, BA.4, BA.5, and BA.2.75 (Fig. 1a).

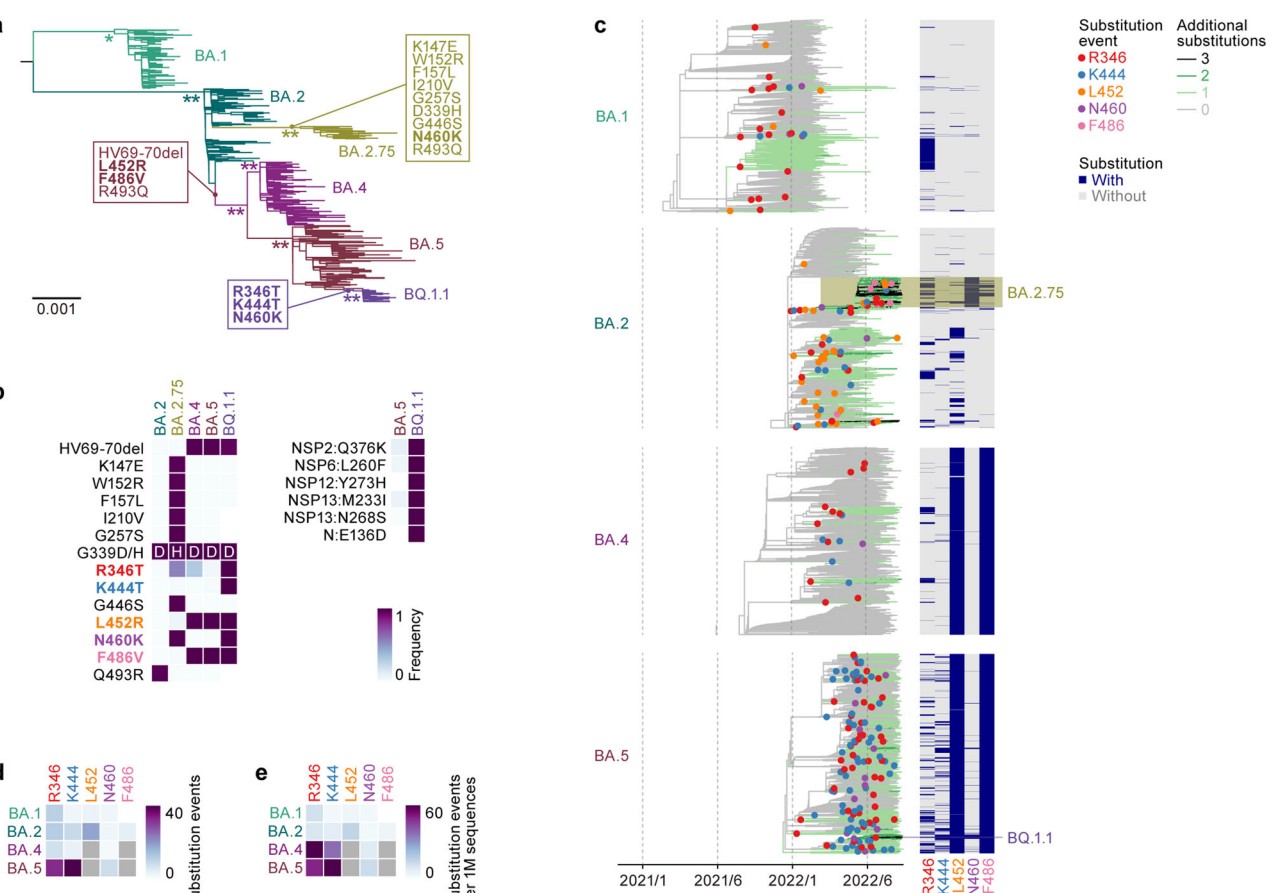

**Fig. 1 | Convergent evolution of Omicron lineages. a** A maximum likelihood (ML) tree of the Omicron lineages. The tree was rooted using an outgroup sequence (B.1.1). The substitutions in the S protein acquired by BA.4/BA.5, BA.2.75, and BQ.1.1 are indicated in the panel, and the five convergent substitutions are indicated in bold. Note that R493Q is a reversion. Bootstrap values, *, ≥0.85; **, ≥0.9. **b** Left, amino acid differences in the S proteins of Omicron lineages. The five convergent substitutions are indicated in bold. Right, amino acid differences in the non-S proteins between BA.5 and BQ.1.1. **c** Left, time-calibrated ML trees for BA.1, BA.2, BA.4, and BA.5. The trees for BA.2 and BA.5 include BA.2.75 and BQ.1.1 lineages,

respectively. The dots indicate estimated substitution events at the convergent sites. The branch color indicates the estimated number of additional substitutions at the convergent sites compared to the most recent common ancestor of each lineage. Right, the substitution profile at the convergent sites. **d, e** The number of substitution events at the convergent sites detected. Raw counts (**d**) and counts per 1 million (M) analyzed sequences (**d**) are shown. Note that L452 and F486 in BA.4/5 are indicated in gray because the common ancestor of BA.4/5 harbors the L452R and F486V substitutions.

As shown in Fig. 1a, BQ.1.1—a latest lineage of concern in November 2022—emerged from the BA.5 cluster. Notably, the substitutions in the S protein, particularly R346X, K444X, L452X, N460X, and F486X, seem to have convergently occurred in a variety of Omicron lineages (hereafter, we refer to these five amino acid residues as *convergent sites* and substitutions at the residues as *convergent substitutions*)[6]. The BQ.1.1 S protein harbors all five of these convergent substitutions (R346T, K444T, L452R, N460K, and F486V) (Fig. 1b, left). Although BQ.1.1 also possesses six non-S substitutions compared to the original BA.5, these substitutions—except for NSP13:N268S—were acquired in lineages ancestral to BQ.1.1 and are not specific to BQ.1.1 (Fig. 1b, right and Supplementary Fig. 1a). To investigate in depth the substitutions at the five convergent sites during Omicron evolution, we constructed phylogenetic trees for BA.1, BA.2 (including BA.2.75), BA.4, and BA.5 (including BQ.1.1) and identified the branches on the trees at which the convergent substitutions occurred (Fig. 1c). The R346 residue showed a higher substitution frequency than that of the other residues in all Omicron lineages (Fig. 1c–e, Supplementary Fig. 1b, c, and Supplementary Table 1). Consistent with our previous study[17], the L452 residue showed the highest substitution frequency in the BA.2 lineage, with 16.8 events per 1M sequence (Fig. 1c–e). Importantly, substitution events were more frequently detected in relatively younger lineages, such as BA.4, BA.5, and BA.2.75, than in the relatively older BA.1 and BA.2 lineages (Fig. 1c–e, Supplementary Fig. 1c). For instance, the R346X and K444X substitutions in BA.4 and BA.5 and the R346X and F486X substitutions in BA.2.75 showed substantially higher substitution frequencies than those in the other lineages (Fig. 1c). The substitution frequencies at R346 (58.5 events per 1M sequence) and K444 (77.6 events per 1M sequence) in BA.5 were approximately 10.4- and 9.4-fold higher than those in BA.2, respectively (Fig. 1e).

## Fitness landscape of the Omicron S protein

We hypothesized that substitutions at these five convergent sites conferred a selective advantage to the circulating viruses, which is why the substitutions were acquired recurrently during the evolution of Omicron. To test this hypothesis, we modeled the relationship between viral epidemic dynamics and S protein substitutions and estimated the effects of all substitutions in the S protein on viral fitness [represented as the relative effective reproduction number ($R_e$)]. Notably, viral fitness or relative $R_e$ could be attributed not only to pure transmissibility but also to immune escape capacity. We classified the viral sequences of Omicron according to the combination of amino acid substitutions in the S protein (referred to as the *S haplotype*) and quantified the daily frequency of each S haplotype-based viral group (Fig. 2a). We analyzed the dataset for 375,121 Omicron sequences collected in the United Kingdom (UK) from March 1, 2022, to October 15, 2022, and classified the sequences into 254 S haplotypes according to the pattern of 107 substitutions in the S protein [or substitution clusters (Supplementary Fig. 1d)]. Inspired by the model established by Obermeyer et al., we subsequently established a Bayesian hierarchal model, which provides the epidemic dynamics of the S haplotype according to relative $R_e$ represented by a linear combination of the effect of substitutions (see Methods). This model can simultaneously estimate i) the effect of each substitution on $R_e$ and (ii) the relative $R_e$ of a viral group represented by each S haplotype. Furthermore, this model can predict the relative $R_e$ for arbitrary sequences of the S protein only based on the profile of substitutions in the S protein (see Methods). The model was fitted to the genome surveillance data from the UK described above. We first investigated the effects of substitutions on $R_e$ (Fig. 2b, Supplementary Fig. 1e, and Supplementary Data 1). The cluster of substitutions specific to BA.1—the earliest Omicron lineage with the lowest $R_e$ overall—and substitution Q493R—acquired once in the common ancestor of Omicron but subsequently lost in BA.5 and BA.2.75—were identified as having negative effects on $R_e$. On

the other hand, substitutions at the five convergent sites were shown to exhibit a clear positive effect on $R_e$ by the model. In particular, the highest positive effects were observed for (i) the L452R and F486V substitutions, acquired by the common ancestor of BA.4 and BA.5; (ii) L452Q, acquired by BA.2.12.1[17]; and (iii) N460K, acquired by BA.2.75 and BQ.1.1 independently. Next, we investigated relative $R_e$ values for viral groups represented by respective S haplotypes (Fig. 2c, Supplementary Fig. 1f, and Supplementary Data 2). We found that S haplotypes with substitutions at the convergent sites, particularly with R346T, K444T, L452R, N460K, and F486V, tended to show higher $R_e$ values. Notably, the S haplotypes corresponding to BQ.1.1, harboring all five convergent substitutions, showed the highest $R_e$ values (1.7-fold higher than that of the major BA.2 haplotype); the second highest was obtained with S haplotypes corresponding to BQ.1, harboring all substitutions apart from R346T.

Next, we assessed the extent to which our model, trained on genome surveillance data from a single country (i.e., the UK), could capture generalizable features about the fitness landscape of Omicron variants. First, we predicted the relative $R_e$ of each S haplotype in each country using the model trained on the UK data. Subsequently, we compared the predicted relative $R_e$ with that estimated by a simple multinomial logistic model based on the data from each country (Fig. 2d, e and Supplementary Fig. 2). The predicted $R_e$ was highly concordant with the estimated $R_e$ (adjusted $R^2 > 0.9$) in most countries investigated. The lower $R^2$ for some countries could be explained by the presence of a few outlier S haplotypes (Supplementary Fig. 2). Furthermore, the trained model successfully predicted the higher $R_e$ of XBB (a recombinant lineage between two highly divergent BA.2 variants[18]) and BA.2.3.20 (a highly diversified BA.2 sublineage, which harbored 10 substitutions compared to BA.2 in the S1 subunit, including L452M, K444R, and N460K[19]), even though these variants were not included in the training data (Fig. 2d). Together, these results suggest that our model captured key information on the fitness landscape underlying the convergent evolution of Omicron lineages observed worldwide in late 2022.

To quantify the total impact of substitutions at the convergent sites on viral fitness, we inferred the proportion of the variation in $R_e$ that can be explained by the substitutions in Omicron lineages. We first calculated the total effect of substitutions at the convergent sites for each S haplotype. Subsequently, we compared this quantity with the relative $R_e$ value for each S haplotype (Fig. 2f). These two quantities were strongly correlated ($R^2 = 0.816$), suggesting that the vast majority (81.6%) of $R_e$ variation in the Omicron lineages can be explained by substitutions at the five convergent sites under our model.

To obtain further insights into the convergent evolution observed in the most recently emerged Omicron lineages, we reconstructed the evolutionary change in viral fitness during BA.5 diversification. The ancestral profile of substitutions in the S protein for each node in the BA.5 tree was reconstructed. Subsequently, we predicted the relative $R_e$ for each ancestral node according to the reconstructed substitution profile using the model above (Fig. 2g). This analysis suggested that the relative $R_e$ was independently elevated in multiple lineages during BA.5 diversification, coupled with substitution events at convergent sites (Fig. 2g, left). Finally, we inferred the evolutionary changes in viral fitness specific to the emergence of BQ.1.1 and revealed that the ancestral lineage of BQ.1.1 acquired the K444T, N460K, and R346T substitutions in this order (Fig. 2g, right)[20]. Importantly, our analysis predicted that the ancestral lineage of BQ.1.1 increased its viral fitness in a stepwise manner, consistent with the acquisitions of these three substitutions (Fig. 2g, right). Taken together, our results suggest that the sublineages descending from BA.5, including BQ.1.1, convergently increased viral fitness by consecutively acquiring substitutions at the R346, N460, and K444 residues.

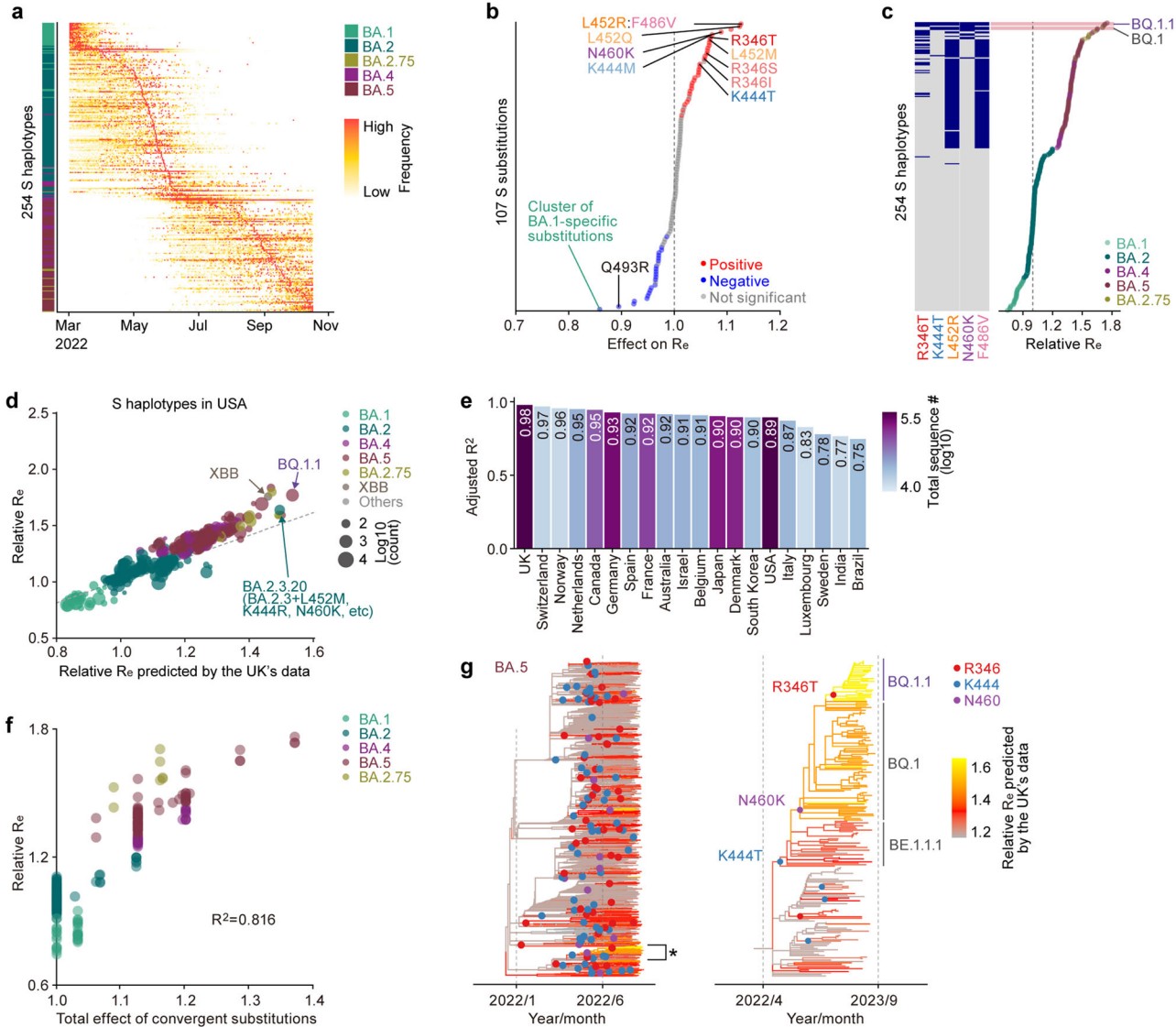

**Fig. 2 | Fitness landscape of S proteins of Omicron lineages as of late 2022.**
**a** Epidemic dynamics of S haplotypes in the UK. Omicron sequences were classified into 254 groups harboring unique sets of substations in the S protein, referred to as S haplotypes. S haplotypes are ordered according to the time of epidemic peak. **b** Effect size of each substitution in the S protein on relative effective reproduction number ($R_e$) estimated by a hierarchal Bayesian model. The posterior mean value is shown. A group of highly co-occurred substitutions (e.g., L452R and F486V) was treated as substitution clusters. The red and blue dots indicate the substitutions with significant positive and negative effects, respectively. The representative substitutions are annotated. **c** Relative $R_e$ value for a viral group represented by each S haplotype, assuming a fixed generation time of 2.1 day. The posterior mean value is shown. The $R_e$ of the major S haplotype in BA.2 is set at 1. The substitution

profile at the five convergent sites is shown on the left. **d** Prediction of the relative $R_e$ of S haplotypes in the USA using the model trained on UK data. The predicted $R_e$ and $R_e$ estimated by a simple multiple logistic model based on USA's data were compared. The dotted line denotes a line with a slope of 1 and an intercept of 0. **e** Adjusted $R^2$ value for the prediction of the $R_e$ of S haplotypes in each country. The bar color indicates the total number of sequences included in the dataset investigated. **f** Comparison between relative $R_e$ and the total effect of substitutions at the convergent sites on $R_e$. Dot indicates a viral group represented by an S haplotype. The dots are colored according to the major classification of the PANGO lineage. **g** Change in viral fitness during BA.5 diversification. The lineages indicated with an asterisk, which includes BQ.1.1, are magnified in the right panel.

## Immune escape of BQ.1.1

It has been recently reported that BQ.1.1 exhibits profound escape from all therapeutic monoclonal antibodies currently approved by the Food and Drug Administration (FDA) in the United States[6,21]. In another study, humoral immunity was elicited by 3-dose treatment with an inactivated vaccine (CoronaVac) and breakthrough infections of prior Omicron subvariants (including BA.1, BA.2, and BA.5) after CoronaVac vaccination, and the results demonstrated that some substitutions detected in BQ.1.1, such as R346T and K444T, contribute to escape from humoral immunity[6]. However, the immune escape ability of BQ.1.1 and the humoral immunity elicited by breakthrough infections of prior Omicron subvariants after mRNA vaccine treatment remain

unaddressed. To experimentally investigate the virological features of BQ.1.1, we first evaluated the immune escape ability of BQ.1.1 using HIV-1-based pseudoviruses. Consistent with our recent study[17], compared to BA.2, BA.5 (2.5-fold) and BQ.1.1 (6.9-fold) more greatly evaded breakthrough BA.2 infection sera, with statistically significant differences (Fig. 3a). Additionally, BQ.1.1 evaded breakthrough BA.2 infection sera more efficiently than BA.5 (2.7-fold, $P = 0.0076$) (Fig. 3a). As shown in Fig. 1a, the BQ.1.1 S protein harbors additional substitutions in the BA.5 S protein, including R346T, K444T, and N460K. To determine the substitution(s) responsible for the immune escape of BQ.1.1 to breakthrough BA.2 infection sera, we prepared BA.5 derivatives bearing each of these three substitutions. However, compared to BA.5,

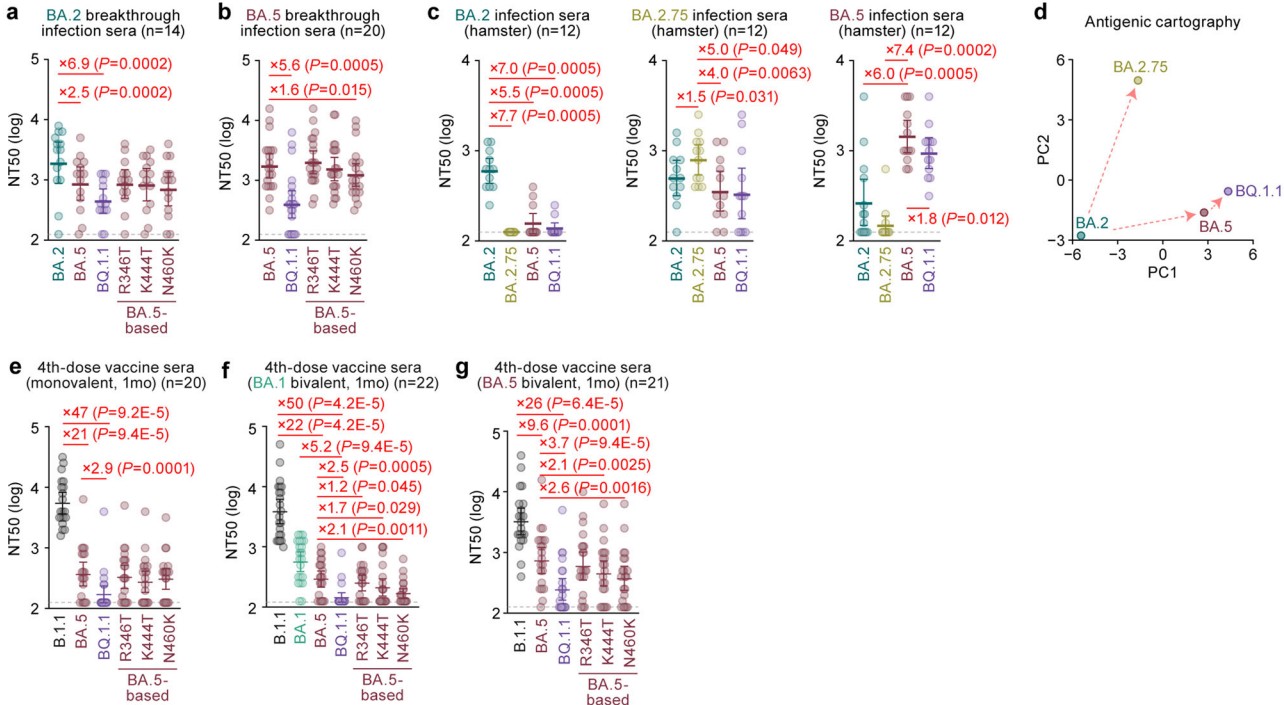

**Fig. 3 | Immune evasion of BQ.1.1.** Neutralization assays were performed with pseudoviruses harboring the S proteins of B.1.1, BA.1, BA.2, BA.2.75, BA.5 and BQ.1.1. The BA.5 S-based derivatives are included in (**a**, **b**, **e**–**g**). The following sera were used. **a**, **b** Convalescent sera from fully vaccinated individuals who had been infected with BA.2 after full vaccination (9 2-dose vaccinated and 5 3-dose vaccinated). 14 donors in total (**a**), and BA.5 after full vaccination (2 2-dose vaccinated donors, 17 3-dose vaccinated donors and 1 4-dose vaccinated donor). 20 donors in total (**b**). **c** Sera from hamsters infected with BA.2 (12 hamsters; left), BA.2.75 (12 hamsters; middle), and BA.5 (12 hamsters; right). **d** Principal component (PC) analysis representing the antigenicity of the S proteins. The analysis is based on the results of neutralization assays using hamster sera (**c**). **e**–**g** 4-dose vaccine sera collected at 1 month (1mo) after the 4-dose monovalent vaccine (19 donors) (**e**), BA.1 bivalent vaccine (22 donors) (**f**), and BA.5 bivalent vaccine (21 donors) (**g**) Assays for each serum sample were performed in triplicate to determine the 50% neutralization titer ($NT_{50}$). Each dot represents one $NT_{50}$ value, and the geometric mean and 95% confidential interval (CI) are shown. Statistically significant differences were determined by two-sided Wilcoxon signed-rank tests. The $P$ values versus BA.2 (**a**, **c**, left), BA.2.75 (**c**, middle), BA.5 (**b**, **c**, right) or B.1.1 (**e**–**g**) are indicated in the panels. The horizontal dashed line indicates the detection limit (120-fold). Information on the convalescent donors is summarized in Supplementary Table 3.

none of the BA.5-based derivatives prepared exhibited resistance to breakthrough BA.2 infection sera (Fig. 3a), suggesting that multiple substitutions cooperatively contribute to the immune escape of BQ.1.1 to breakthrough BA.2 infection sera. For breakthrough BA.5 infection sera, BQ.1.1 more efficiently evaded breakthrough BA.5 infection sera than BA.5 with a statistically significant difference (5.6-fold, $P < 0.0001$) (Fig. 3b). Importantly, the breakthrough BA.5 infection sera obtained from six individuals (five breakthrough infection cases after 3-dose vaccination and a breakthrough infection case after 2-dose vaccination) did not exhibit a neutralizing effect against BQ.1.1 in this experimental setup. We then assessed the substitutions that confer the ability to evade breakthrough BA.5 infection sera. The N460K substitution conferred significant escape from breakthrough BA.5 infection sera (1.6-fold, $P = 0.016$), while the other two substitutions did not affect the immune escape from breakthrough BA.5 infection sera (Fig. 3b). Compared to the immune escape of BQ.1.1 to breakthrough BA.5 infection sera (5.6-fold), the immune evasion acquired by the N460K substitution (1.6-fold) is less robust (Fig. 3b). Therefore, the results suggest that the immune escape ability of BQ.1.1 from breakthrough BA.5 infection sera can be attributed to multiple substitutions in the receptor binding domain (RBD) of BQ.1.1 S (R346T, K444T, and N460K), similar to breakthrough BA.2 infection sera.

To further address the difference in antigenicity among Omicron subvariants, including BQ.1.1, we used sera obtained from infected hamsters at 16 days post-infection (d.p.i.). Consistent with our previous studies[2,17], the hamster sera infected with BA.2, BA.5, or BA.2.75 most efficiently showed neutralization activity against the variant of

virus infected, while these sera were less or not cross-reactive against the other variants (Fig. 3c). In the case of BA.5 infection sera, BQ.1.1 was 1.8-fold more efficient than BA.5 in evading neutralization (Fig. 3c). To depict the difference in antigenicity among BA.2, BA.5, BA.2.75 and BQ.1.1, we further analyzed the neutralization dataset of hamster sera shown in Fig. 3c. As shown in Fig. 3d, the cross-reactivity of each Omicron subvariant was well concordant to their phylogenetic relationship (Fig. 1a), and the antigenicity of BQ.1.1 is relatively more similar to BA.5 than BA.2 and BA.2.75. Nevertheless, BQ.1.1 achieved profound escape from the humoral immunity induced by BA.5 breakthrough infection (Fig. 3b). We then assessed the sensitivity of BQ.1.1 to the 4-dose vaccine sera. As shown in Fig. 3e-g, BQ.1.1 significantly escaped from monovalent vaccine sera (47-fold, $P < 0.0001$), BA.1 bivalent vaccine sera (50-fold, $P < 0.0001$), and BA.5 bivalent vaccine sera (26-fold, $P < 0.0001$) compared with B.1.1. Although the three BA.5 derivatives were comparable with BA.5 against monovalent vaccine sera (Fig. 3e), at least two substitutions (K444T and N460K) contribute to escape from BA.1 or BA.5 bivalent vaccine sera (Fig. 3f, g). These observations suggest that the three substitutions in BQ.1.1 S protein (R346T, K444T and N460K) are critical and specific for evading BA.5 infection-induced herd immunity in the human population.

## ACE2 recognition of BQ.1.1 S protein
We then evaluated the features of the BQ.1.1 S protein that potentially affect viral infection and replication. Yeast surface display assay[2,17,22–26] showed that the dissociation constant ($K_D$) value of BQ.1.1 RBD (0.66 ± 0.11) to the human ACE2 molecule is significantly lower than

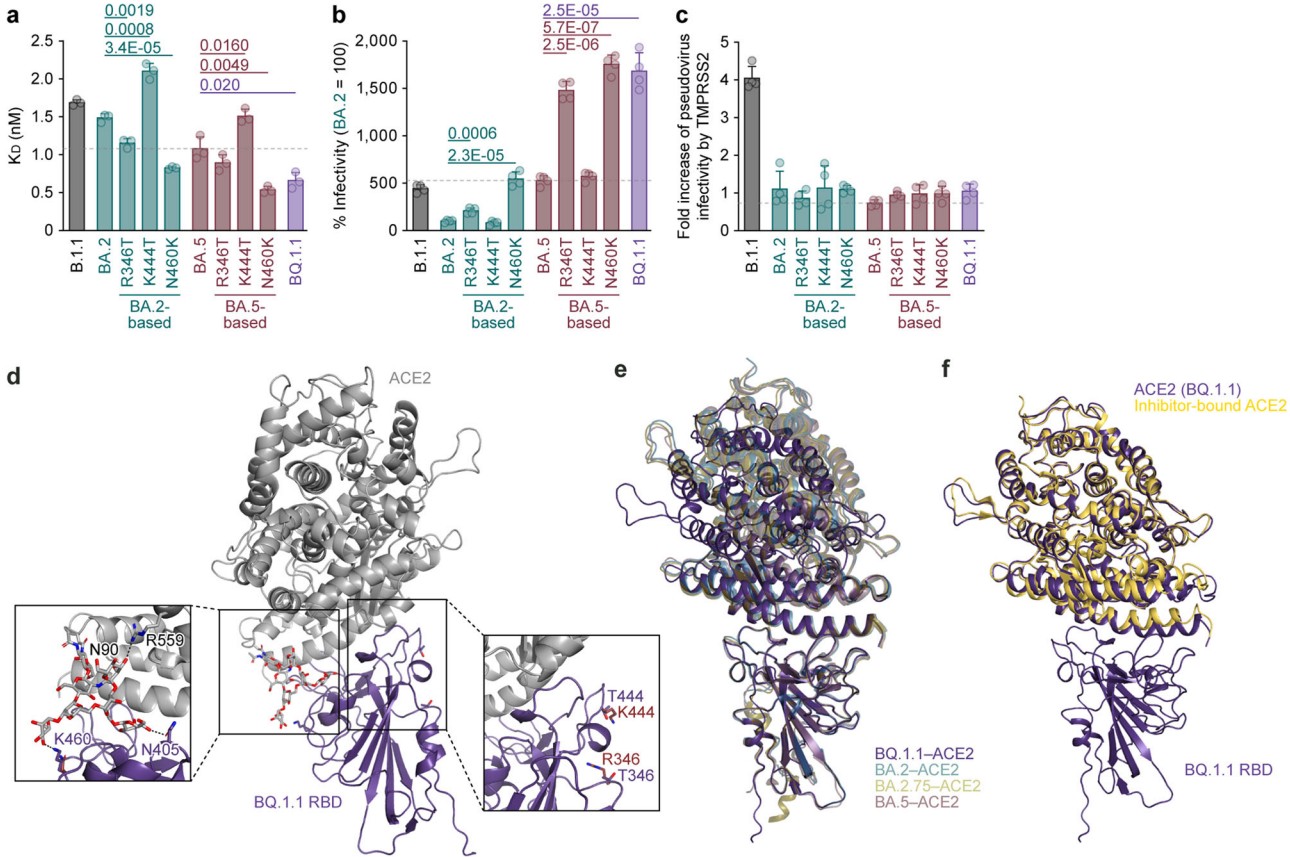

**Fig. 4 | Interaction between BQ.1.1 S and ACE2. a** Binding affinity of the receptor binding domain (RBD) of the SARS-CoV-2 S protein to ACE2 by yeast surface display. The dissociation constant ($K_D$) value indicating the binding affinity of the RBD of the SARS-CoV-2 S protein to soluble ACE2 when expressed on yeast is shown. **b** Pseudovirus assay. HOS-ACE2/TMPRSS2 cells were infected with pseudoviruses bearing each S protein. The amount of input virus was normalized based on the amount of HIV-1 p24 capsid protein. The percent infectivity compared to that of the virus pseudotyped with the BA.2 S protein are shown. **c,** Fold increase in pseudovirus infectivity based on TMPRSS2 expression. **d–f** The BQ.1.1 RBD bound to ACE2 trapped in the closed conformation. **d** Crystal structure of the BQ.1.1 RBD-human ACE2 complex. Characteristic substitutions in the BQ.1.1 RBD and an N-linked glycan on N90 of human ACE2 are shown in purple and gray sticks. In the close-up view, corresponding residues in the BA.4/5 RBD-human ACE2 complex structure (PDB: 7XWA)[17] are also shown in brown sticks. The BQ.1.1 RBD and ACE2 residues

recognizing the glycan are shown in stick representation. Dashed lines represent hydrogen bonds. **e** Superimposition of the BQ.1.1 RBD-human ACE2 complex structure (purple) onto previously reported structures of SARS-CoV-2 RBD bound to human ACE2. BQ.1.1, purple; BA.2[27], pale green, PDB: 7ZF7; BA.2.75[30], khaki, PDB: 8ASY; BA.5[17], brown, PDB: 7XWA. BA.2, BA.2.75. BA.5 are shown as transparent. **f** Superimposition of the BQ.1.1 RBD-human ACE2 complex structure (purple) onto a previously reported structure of an inhibitor bound human ACE2 (pale yellow, PDB: 1R4L)[28]. Assays were performed in triplicate (**a**) or quadruplicate (**b**). The presented data are expressed as the average ± standard deviation (SD) (**a–c**). Each dot indicates the result of an individual replicate. The dashed horizontal lines indicate the value of BA.5. Statistically significant differences versus each parental S protein and those between BA.5 and BQ.1.1 were determined by two-sided Student's *t* tests.

that of the parental BA.5 RBD (1.08 ± 0.16) (Fig. 4a), suggesting that BQ.1.1 increased the binding affinity to human ACE2 during evolution from BA.5. To determine the responsible substitutions in the BQ.1.1 S protein that enhance ACE2 binding affinity, we prepared the RBDs of BA.2 and BA.5 S proteins that possess a BQ.1.1-specific substitution different from parental BA.5 (i.e., R346T, K444T and N460K). Consistent with our recent study[2], the N460K substitution significantly increased the binding affinity of the S proteins of BA.2 and BA.5 to human ACE2 (Fig. 4a). On the other hand, the K444 substitution significantly decreased ACE2 binding affinity regardless of the backbone of the S protein (Fig. 4a). The R346T substitution increased the ACE2 binding affinity of BA.2 RBD but not that of BA.5 RBD (Fig. 4a). The in vitro observations using yeasts (Fig. 4a) were then verified by using an HIV-1-based pseudovirus system. As shown in Fig. 4b, the infectivity of the BQ.1.1 pseudovirus was significantly higher than that of the BA.2 (17-fold) and BA.5 (3.2-fold) pseudoviruses. In our recent study[17], at least three mutations were detected in BA.5 S protein (compared to BA.2 S protein), HV69-70del, L452R, and F486V contribute to the increase in pseudovirus infectivity. When we particularly focused on

the three additional mutations detected in the BQ.1.1 S protein compared to the BA.5 S protein, R346T, K444T, and N460K, we found that R346T and N460K but not K444T significantly increased pseudovirus infectivity, and this effect was independent of the backbone of the S protein (Fig. 4b). To assess the association of TMPRSS2 usage with the increased pseudovirus infectivity of BQ.1.1, we used HEK293-ACE2/TMPRSS2 cells and HEK293-ACE2 cells, on which endogenous surface TMPRSS2 is undetectable[26], as target cells. As shown in Fig. 4c, the infectivity of the BQ.1.1 pseudovirus was not increased by TMPRSS2 expression, suggesting that TMPRSS2 is not associated with an increase in the infectivity of the BQ.1.1 pseudovirus.

To gain structural insight into ACE2-receptor recognition by the BQ.1.1 RBD, we performed X-ray crystallographic analysis of the BQ.1.1 RBD-human ACE2 complex and determined its structure at a resolution of 2.78 Å (Fig. 4d–f and Supplementary Table 2). Three amino acid substitutions (R346T, K444T, and N460K) in the BQ.1.1 RBD different from the BA.4/5 RBD were focused on the interaction with human ACE2 (Fig. 4d). First, the interaction between N460K in BQ.1.1 RBD and the N-linked glycan on N90 of human ACE2 that we observed in our

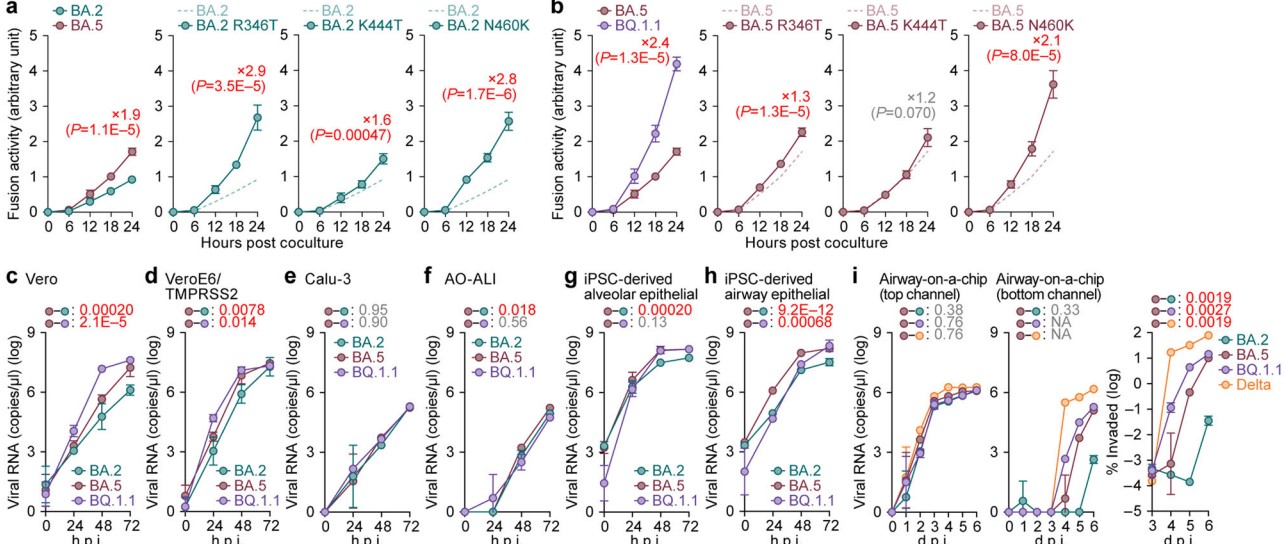

**Fig. 5 | Virological characteristics of BQ.1.1 in vitro. a, b** S-based fusion assay in Calu-3 cells. The recorded fusion activity (arbitrary units) is shown. The dashed green line (**a**) and the dashed brown line (**b**) indicate the results of BA.2 and BA.5, respectively. The red number in each panel indicates the fold difference between BA.2 (**a**) or BA.5 (**b**) and the derivative tested at 24 h post coculture. **c–i** Growth kinetics of BQ.1.1. Clinical isolates of BA.2, BA.5, BQ.1.1 and Delta (only in **i**) were inoculated into Vero cells (**c**), VeroE6/TMPRSS2 cells (**d**), Calu-3 cells (**e**), the human airway organoid-derived air-liquid interface (AO-ALI) system (**f**), human induced pluripotent stem cell (iPSC)-derived alveolar epithelial cells (**g**), iPSC-derived airway epithelial cells (**h**), and an airway-on-a-chip system (**i**). h.p.i., hours post-infection; d.p.i., days post-infection. The copy numbers of viral RNA in the culture supernatant (**c–e**), the apical sides of cultures (**f–i**), and the top (**i**, left) and bottom (**i**, middle) channels of an airway-on-a-chip were routinely quantified by RT−qPCR. In (**i**, right), the percentage of viral RNA load in the bottom channel per top channel during 3−6 d.p.i. (i.e., the % invaded virus from the top channel to the bottom channel) is shown. Assays were performed in triplicate (**i**) or quadruplicate (**a–h**). The presented data are expressed as the average ± standard deviation (SD) (**a**, **b**) or standard error of mean (SEM) (**c–i**). Statistically significant differences versus BA.2 (**a**) and BA.5 (**b–i**) across timepoints were determined by multiple regression. The familywise error rates (FWERs) calculated using the Holm method are indicated in the figures. NA, not applicable.

recent study focusing on BA.2.75[2] was verified in this study. Surprisingly, the N405 in BQ.1.1 RBD and ACE2 R599 were also involved in the interaction with the glycan on N90 of ACE2, despite not being detected in the ACE2 complex structure in BA.4/5[17]. As a result, three amino acid residues, BQ.1.1 RBD K460 and N405 and ACE2 N599 formed an interaction network across the glycan on N90 of ACE2. The formation of this interaction network is consistent with the enhanced affinity of the BQ.1.1 RBD against ACE2 and increased infectivity caused by N460K (Fig. 4a, b). In fact, while the electron density of only a small part of the N-linked glycan on N90 of ACE2 was observed in the BA.4/5 RBD-human ACE2 complex structure, the electron density of the corresponding glycan was well observed in the BQ.1.1 RBD-human ACE2 complex structure (Supplementary Fig. 3a, b). Next, regarding R346T and K444T, these amino acids in BQ.1.1 were not directly involved in the interaction with ACE2 in BQ.1.1 RBD-human ACE2 complex structure, which is the same in other previous variants[17,26,27]. R346T and K444T are both located on the loop structure and are replaced with less-bulky threonine (T) residues, but compared to the BA.5-human ACE2 complex structure, no changes in the loop structure or the orientation of the surrounding amino acid residues were observed.

Finally, it should be noted that ACE2 bound to BQ.1.1 RBD is in a closed conformation (Fig. 4e, f). Structures of the closed ACE2 have been reported as an inhibitor-bound form[28] (Fig. 4f) and as a high-affinity ACE2 mutant bound to B.1.1 RBD[29]. In contrast, ACE2 with an open confirmation has been observed in complex structures with BA.2, 4/5, 2.75 RBDs (Fig. 4e)[17,26,27]. As mentioned above, except for the interaction network among BQ.1.1 RBD K460, N405, and ACE2 R599 via the glycans on N90 of ACE2, the interaction of the receptor-binding motif of the BQ.1.1 S protein with ACE2 is consistent with previous reports even in the closed ACE2 structure[2,27,30]. Therefore, the interaction network via the glycans on N90 of ACE2 observed in this study may prefer to bind with the closed ACE2. In the closed ACE2 structure,

an unidentified substrate-like electron density was observed around the catalytic site, and the location of this density is identical to that of an inhibitor against ACE2[28] (Supplementary Fig. 3c, d). A closed conformation has also been reported for an ACE2 mutant with increased affinity to the SARS-CoV-2 RBD[29], and similarly, an unidentified substrate-like electron density has been observed at the same position. Therefore, it may be conceivable that closed ACE2, which incorporates a host-derived substrate/peptide during protein expression, can bind to RBD in some combinations.

## Fusogenicity of BQ.1.1 S

The fusogenicity of BQ.1.1 S protein was measured by a fusion assay based on the SARS-CoV-2 S protein[2,17,22,26,31–33] using Calu-3 cells. The surface expression level of the BQ.1.1 S protein was significantly lower than that of BA.2, but the BQ.1.1 and BA.5 expression levels were comparable (Supplementary Fig. 3e). In the BA.2 S derivatives, R346T and N460K, significantly decreased surface expression (Fig. 5a). In the BA.5 S derivatives, N460K significantly decreased surface expression, while K444T increased surface expression (Supplementary Fig. 3e). The fusogenicity of BA.5 S protein was greater than that of BA.2 S protein (Fig. 5a), which is consistent with our recent studies[2,17]. More importantly, compared to the BA.5 S protein, the BQ.1.1 S protein was significantly more fusogenic (Fig. 5b). Additional experiments using the S derivatives based on BA.2 and BA.5 showed that the R346T and N460K substitutions significantly increased the S-mediated fusogenicity independently of the S backbone (Fig. 5a, b). Together with our recent studies[2], the results indicated that the N460K substitution, which is detected in BA.2.75, increased ACE2 binding affinity (i.e., decrease in the $K_D$ value) (Fig. 4a), increased pseudovirus infectivity (Fig. 4b) and S-mediated fusogenicity (Fig. 5a, b). Interestingly, the R346T substitution also significantly increased ACE2 binding affinity and S-based fusogenicity, while the K444T substitution negatively

affected these experimental parameters (Figs. 4b, c and 5a, b). These results suggest that, compared to BA.5, the virological features of BQ.1.1 S protein, including increased ACE2 binding affinity, pseudovirus infectivity and fusogenicity, are attributed to the R346T and N460K substitutions.

## Growth kinetics of BQ.1.1 in vitro

To investigate the growth kinetics of BQ.1.1 in vitro cell culture systems, we inoculated clinical isolates of BA.2[26], BA.5[2] and BQ.1.1 into multiple cell cultures. The growth of BQ.1.1 in Vero cells (Fig. 5c) and VeroE6/TMPRSS2 cells (Fig. 5d) was significantly greater than that of BA.5, and the growth of BQ.1.1 and BA.5 was comparable in Calu-3 cells (Fig. 5e), the human airway organoid-derived air-liquid interface (AO-ALI) system (Fig. 5f), and human induced pluripotent stem cell (iPSC)-derived alveolar epithelial cells (Fig. 5g). However, the growth of BQ.1.1 in iPSC-derived airway epithelial cells was significantly lower than that of BA.5 (Fig. 5h).

The differences in replication kinetics between the AO-ALI system (Fig. 5f) and iPSC-derived airway epithelial cells (Fig. 5h) are likely due to the differences in the characteristics of these cells. AO-ALI cells are differentiated from normal human bronchial epithelial cells, while iPSC-derived airway epithelial cells are differentiated from iPS cells. Therefore, it is expected that the maturity and cell population of AO-ALI are different from those of iPSC-derived airway epithelial cells, and these differences impact viral growth kinetics.

To evaluate the impact of BQ.1.1 infection on the airway epithelial and endothelial barriers, we used an airway-on-a-chip system[2,34,35]. By measuring the amount of virus that invaded from the top channel (Fig. 5i, left) to the bottom channel (Fig. 5i, middle), we evaluated the ability of viruses to disrupt the airway epithelial and endothelial barriers. Weakening of SARS-CoV-2 infection-mediated airway epithelial and endothelial barriers can also be confirmed by the disruption of VE-cadherin−mediated adherens junctions[34]. Notably, the percentage of virus that invaded the bottom channel of the BQ.1.1-infected airway-on-a-chip was significantly higher than that of the BA.5-infected airway-on-a-chip (Fig. 5i, right). Together with the findings of the S-based fusion assay (Fig. 5b), these results suggest that BQ.1.1 exhibits higher fusogenicity than that of BA.5.

## Virological characteristics of BQ.1.1 in vivo

To investigate the virological features of BQ.1.1 in vivo, clinical isolates of Delta[32], BA.5[2], and BQ.1.1 [10,000 50% tissue culture infectious dose (TCID$_{50}$)] were intranasally inoculated into hamsters under anesthesia. Consistent with our previous studies[2,32], Delta infection resulted in weight loss (Fig. 6a, left). On the other hand, the body weights of BA.5- and BQ.1.1-infected hamsters did not increase compared with that of the negative control and were relatively similar (Fig. 6a, left). We then analyzed the pulmonary function of infected hamsters as reflected by two parameters, enhanced pause (Penh) and the ratio of time to peak expiratory flow relative to the total expiratory time (Rpef). Among the four groups, Delta infection resulted in significant differences in these two respiratory parameters compared to BA.5 (Fig. 6a, middle and right), suggesting that Delta is more pathogenic than BA.5. In contrast, the Penh value of BQ.1.1-infected hamsters was significantly lower than that of BA.5-infected hamsters, and the Rpef value of BQ.1.1-infected hamsters was significantly higher than that of BA.5-infected hamsters (Fig. 6a, middle and right). These observations suggest that the pathogenicity of BQ.1.1 is similar to or even less than that of BA.5.

To evaluate viral spread in infected hamsters, we routinely measured the viral RNA load in oral swabs. Although the viral RNA loads of the hamsters infected with Delta were significantly higher than those infected with BA.5, there was no significant difference between BQ.1.1 and BA.5 (Fig. 6b, left). To address the possibility that BQ.1.1 more efficiently spreads in the respiratory tissues, we collected the lungs of infected hamsters at 2 and 5 d.p.i., and the collected tissues were

separated into the hilum and periphery regions. However, the viral RNA loads in both the lung hilum and periphery of BA.5-infected hamsters were comparable to those of Delta- and BQ.1.1-infected hamsters (Fig. 6b, middle and right), suggesting that the viral dissemination of BQ.1.1 in the lungs is comparable to BA.5. To further investigate viral spread in the respiratory tissues of infected hamsters, we performed immunohistochemical (IHC) analysis targeting the viral nucleocapsid (N) protein. Similar to our previous studies[17,26,31], epithelial cells in the upper tracheae of infected hamsters were sporadically positive for viral N protein at 2 d.p.i., but there were no significant differences among the three viruses, including BQ.1.1 (Supplementary Fig. 4a), although tracheal inflammation tended to remain in BQ.1.1-infected hamsters. In the alveolar space around the bronchi/bronchioles at 2 d.p.i., N-positive cells were detected in Delta-infected hamsters (Fig. 6c, left and Supplementary Fig. 4b). In contrast, the percentage of N-positive cells in the lungs of BQ.1.1- and BA.5-infected hamsters was relatively low and comparable (Fig. 6c, left and Supplementary Fig. 4b). At 5 d.p.i., N-positive cells were detected in the peripheral alveolar space in Delta-infected hamsters, while the N-positive areas of BQ.1.1- and BA.5-infected hamsters were sporadic and faintly detectable (Fig. 6c, right and Supplementary Fig. 4b). These data suggest that the spreading efficiency of BQ.1.1 in the lungs of infected hamsters is comparable to that of BA.5.

## Intrinsic pathogenicity of BQ.1.1

To investigate the intrinsic pathogenicity of BQ.1.1, we analyzed the formalin-fixed right lungs of infected hamsters at 2 and 5 d.p.i. by carefully identifying the four lobules and main bronchus and lobar bronchi sectioning each lobe along with the bronchial branches (Fig. 6d). Histopathological scoring was performed according to the criteria described in our previous studies[32]. Consistent with our previous studies[2,31,32], all five histological parameters as well as the total score of the Delta-infected hamsters were significantly greater than those of the BA.5-infected hamsters (Fig. 6e). When we compared the histopathological scores of Omicron subvariants, total histopathological scores were comparable between BQ.1.1-infected hamsters and BA.5-infected hamsters with some enhancement in bronchitis/bronchiolitis at 2 d.p.i. and presence of type II pneumocytes at 5 d.p.i. of BQ.1.1 (Fig. 6e). Altogether, these histopathological analyses suggest that the intrinsic pathogenicity of BQ.1.1 is lower than that of Delta and comparable to that of BA.5.

## Discussion

Here, we illuminated the convergent evolution of Omicron that has led to the emergence of recent Omicron subvariants of concern, BQ.1.1. Our phylogenetic and modeling analyses showed that substitutions at five sites of the S protein, including R346, K444, and N460, have been convergently acquired and are associated with increasing $R_e$ (Figs. 1, 2). We also characterized the effect of the R346T, K444T, and N460K substitutions present in BQ.1.1 on viral properties. These substitutions are involved in the escape from BA.2/BA.5 infection sera and 4-dose vaccine sera, but each substitution alone cannot confer this viral property (Fig. 3). Importantly, BQ.1.1−harboring five convergent substitutions−possesses increased ACE2 binding affinity, pseudovirus infectivity, and fusogenicity, which is attributed to substitutions R346T and N460K (Figs. 4 and 5). However, acquiring these substitutions did not increase the pathogenicity of BQ.1.1 compared to BA.5 in a hamster model (Fig. 6).

In this study, we quantified independent substitution events at the five convergent sites and showed that these substitutions were frequently and recurrently acquired in Omicron lineages, particularly in younger lineages, such as BA.4, BA.5, and BA.2.75 (Fig. 1c−e). Convergent substitution in the S protein has previously been documented for substitution N501Y in early SARS-CoV-2 VOCs[36]. However, the extent of convergence and number of convergent sites we

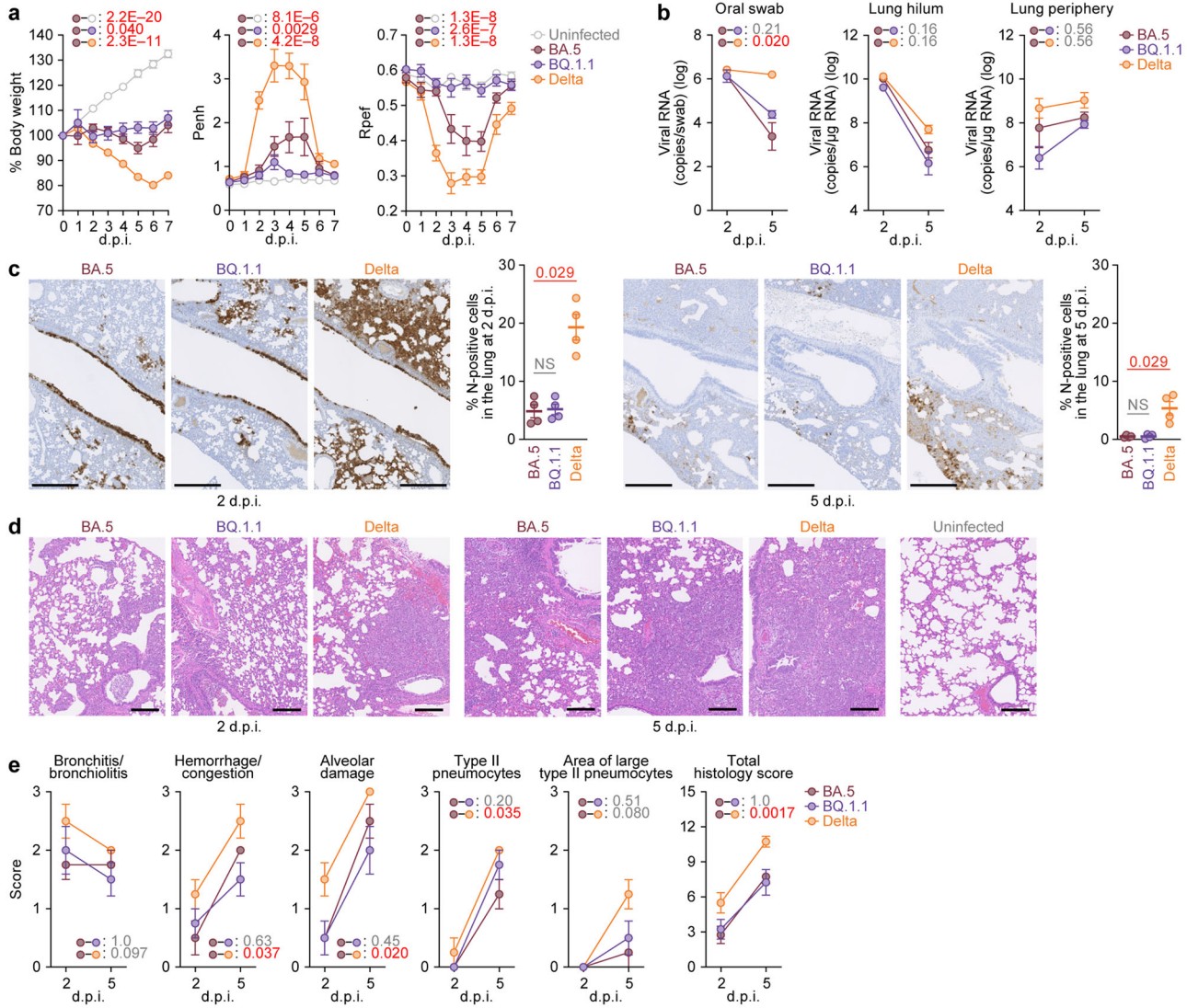

**Fig. 6 | Virological characteristics of BQ.1.1 in vivo.** Syrian hamsters were intra-nasally inoculated with BA.5, BQ.1.1 and Delta. Six hamsters of the same age were intranasally inoculated with saline (uninfected). Six hamsters per group were used to routinely measure the respective parameters (**a**). Four hamsters per group were euthanized at 2 and 5 days post-infection (d.p.i.) and used for virological and pathological analysis (**b–e**). **a** Body weight, enhanced pause (Penh), and the ratio of time to peak expiratory flow relative to the total expiratory time (Rpef) values of infected hamsters ($n = 6$ per infection group). **b** (Left) Viral RNA loads in the oral swab ($n = 6$ per infection group). (Middle and right) Viral RNA loads in the lung hilum (middle) and lung periphery (right) of infected hamsters ($n = 4$ per infection group). **c** Immunohistochemical (IHC) analysis of the viral N protein in the lungs at 2 d.p.i. (left) and 5 d.p.i. (right) of infected hamsters. Representative Figures (N-positive cells are shown in brown) and the percentage of N-positive cells in whole lung lobes ($n = 4$ per infection group) are shown. NS, not significant. The raw data

are shown in Supplementary Fig. 4b. **d, e** Haematoxylin and eosin (H&E) staining of the lungs of infected hamsters. Representative figures are shown in (**d**). Uninfected lung alveolar space and bronchioles are also shown. **e** Histopathological scoring of lung lesions ($n = 4$ per infection group). Representative pathological features are reported in our previous studies[2,17,26,31,32]. In (**a–c,e**), data are presented as the average ± standard error of mean (SEM). In (**c**), each dot indicates the result of an individual hamster. In (**a, b**), and (**e**), statistically significant differences between BA.5 and other variants across timepoints were determined by multiple regression. In (**a**), the 0 d.p.i. data were excluded from the analyses. The familywise error rates (FWERs) calculated using the Holm method are indicated in the figures. In (**c**), the statistically significant differences between BA.5 and other variants were determined by a two-sided Mann–Whitney $U$ test. In (**c, d**), each panel shows a representative result from an individual infected hamster. Scale bars, 500 μm (**c**); 200 μm (**d**).

demonstrate here are much larger. Furthermore, to determine the fitness landscape of the S protein underlying convergent evolution in late 2022, we established a statistical model that predicts viral fitness according to the S protein substitution profile (Fig. 2). Our model, trained on the UK sequencing data, successfully predicted the fitness of variants in other countries and variants absent in the training data, suggesting that the model captures important information about the fitness landscape of the Omicron S protein (Fig. 2d, e). Our analysis shows that the five convergent substitutions significantly increase $R_e$, and the majority of the estimated $R_e$ variation within Omicron in the UK can be explained by substitutions at the five convergent sites,

suggesting that the evolution of Omicron may follow a simple evolutionary rule (Fig. 2b, f). Moreover, our analysis suggests that a variety of BA.5 subvariants have convergently increased their viral fitness, and the stepwise acquisition of K444T, N460K, and R346T led to the emergence of BQ.1.1, which showed the highest $R_e$ in the analyzed sequence dataset (Fig. 2g). Together, our analyses highlight how convergent substitutions at five S protein sites have progressively increased the viral fitness of competing, cocirculating variants, leading to the "variant soup" observed in late 2022.

Viral fitness could be attributed not only to pure transmissibility but also to capacity of immune escape. In particular, since human

populations are imprinted with immunity by vaccinations and natural infections as of 2022, viral fitness is strongly affected by selective pressure from humoral immunity[16]. Indeed, we demonstrated that BQ.1.1, which harbors five convergent substitutions and showed the highest viral fitness among the variants we investigated, is more resistant to the humoral immunity induced by BA.2 and BA.5 break-through infections and 4-dose vaccination than BA.5. Since substitutions at R346, K444, and F486 are associated with escape from humoral immunity and monoclonal antibodies[5,6,17], these substitutions increase viral fitness, likely by increasing immune escape capacity. On the other hand, L452R and N460K increase the binding ability of the S protein to human ACE2 and the infectivity of pseudoviruses[2,17]. Therefore, although it should be noted that i) these viral properties are not direct evidence to evaluate pure viral transmissibility and ii) the properties conferred by each substitution may not be exclusive, these substitutions increased viral fitness likely by increasing the viral properties described above.

There are two possible explanations for the accumulated amino acid substitutions at the convergent sites in relatively younger Omicron lineages, such as BA.4, BA.5 and BA.2.75. One possibility is the epistasis among amino acid substitutions; the fitness of a substitution differs depending on the presence of the other substitutions and/or the backbone sequence (similar to how the original Omicron genotype likely emerged[37]). In our previous studies, the L452R substitution in the BA.4/5 S protein[17] and the N460K substitution in the BA.2.75 S protein[2] increase their binding ability to human ACE2. More importantly, we showed that these substitutions could compensate for the negative effects of the other substitutions that contribute to evasion from humoral immunity but decrease ACE2 binding ability; for BA.4/5, L452R compensates for the attenuated ACE2 binding affinity by the F486V substitution[4,7–9], while for BA.2.75, N460K compensates for the attenuated ACE2 binding affinity by the G446S substitution[2,7,11]. Acquiring substitutions that potentially increase ACE2 binding ability, such as L452R and N460K, may be a factor that increase substitution frequency at the convergent sites in BA.4, BA.5, and BA.2.75. Another possibility is that the effect of substitutions on viral fitness can change over time due to changes in immune selective pressures in the human population by vaccinations and/or natural infections with a variety of SARS-CoV-2 variants. These two possibilities are not mutually exclusive, and these two factors could contribute to accelerated substitutions at convergent sites.

Through structural analysis, we revealed the mechanism for the enhanced binding between the BQ.1.1 S protein and human ACE2 resulting from the N460K substitution in the S protein that has remained in the recent Omicron lineage. In a previous report, we suggested that RBD K460 might interact with the N-linked glycan on N90 of ACE2 based on the cryo-electron microscopy structure of BA.2.75 S protein trimer in complex with ACE2, but the interaction was not visible due to the low resolution of the RBD-ACE2 binding site[2]. Fortunately, the interaction of RBD K460 with the glycan on N90 of ACE2 could be clearly observed in the crystal structure of the BQ.1.1 RBD bound to human ACE2. Through the interaction network formed by BQ.1.1 RBD K460 and N405 as well as ACE2 R599 via the N-linked glycan on ACE2 N90, the BQ.1.1 S protein exhibited a unique binding mode, in which the S protein bound to the closed conformation of ACE2. This binding mode might be related to the findings that BQ.1.1 exhibits a higher affinity for ACE2 and membrane fusion ability than that of BA.5.

Our previous studies focusing on Delta[32], Omicron BA.1[31], BA.5[17], and BA.2.75[2] showed that the intrinsic pathogenicity of SARS-CoV-2 variants is closely related to the fusogenicity of viral S proteins. Therefore, the observations showing the higher fusogenicity of BQ.1.1 S protein than the BA.5 S protein based on the S-based fusion assay (Fig. 5b) and the experiments using airway-on-a-chip (Fig. 5i) suggest that the intrinsic pathogenicity of BQ.1.1 is increased

compared to that of BA.5. However, it was unexpected that the intrinsic pathogenicity of BQ.1.1 in a hamster model was comparable to or even lower than that of BA.5 (Fig. 6). This discrepancy between viral fusogenicity and viral intrinsic pathogenicity is reminiscent of the previous two studies on Omicron BA.2. We first showed that compared to the BA.1 S protein, the BA.2 S protein is more fusogenic[26]. We then artificially generated a BA.2 S-bearing recombinant SARS-CoV-2, in which the non-S region of the viral genome is derived from ancestral SARS-CoV-2, and demonstrated that the BA.2 S-bearing virus is more pathogenic than BA.1 S-bearing virus in hamsters[26]. On the other hand, Uraki et al. showed that the intrinsic pathogenicity of clinical BA.2 isolates is comparable to that of clinical BA.1 isolates[38]. Because the difference between our study[26] and others[38] is likely explained by the viral genome sequence in the non-S region, it is suggested that the BA.2 S protein bears the potential to exhibit augmented pathogenicity when compared to the BA.1 S protein, whereas mutations in non-S regions of the BA.2 genome potentially attenuate viral pathogenicity. In fact, a recent study showed that non-S substitutions in Omicron could attenuate its intrinsic pathogenicity[39], and we found at least six substitutions in the non-S region of BQ.1.1 when compared to that of BA.5. Therefore, it is possible that there are factors other than the S protein that modulate intrinsic viral pathogenicity.

Notably, the estimation of viral fitness performed in the present study involves several limitations. First, the estimation of viral fitness from genome surveillance data is subject to uncertainty, primarily due to biased and/or incomplete sampling of circulating variants. Second, the statistical model used in the present study is simple (but inter-pretable) so that this model does not consider the epistatic effects between substitutions or substitutions absent in the training data, similar to the model by Obermeyer et al.[40]. Most likely due to the simplicity of the model, our model tends to slightly underestimate the relative $R_e$ of variants with higher $R_e$ (Supplementary Fig. 2). Third, the viral fitness estimated by our model may not be generalizable to different situations since the fitness landscape can change depending on environmental factors, particularly the level of immunity in human populations. Our model was trained on the surveillance data for a specific period (i.e., from April 1, 2022 to October 15, 2022) in a specific region (i.e., the UK). This is the biggest difference between our model and the model in Obermeyer et al., which utilizes an additional layer to incorporate data from multiple regions. Nevertheless, our model successfully predicted the fitness of i) variants in countries other than the UK and ii) variants not included in the training data (e.g., XBB), suggesting that it could capture the key features of the fitness land-scape of Omicron lineages as of late 2022 (Fig. 2d, e). These results also imply that the fitness landscape of Omicron lineages may not sub-stantially differ between countries at least as of 2022. It should, how-ever, be mentioned that the results represent the landscape experienced by the virus during the analyzed period (mid to late 2022), which is likely to change with adapting population immunity.

Altogether, we have illuminated the evolutionary principles behind the convergent evolution of Omicron as of late 2022 and elu-cidated the properties of BQ.1.1, a new variant that emerged due to convergent evolution. Knowledge on SARS-CoV-2 evolution provides opportunities to the development of new methods for predicting future epidemic variants and the early detection of highly transmis-sible variants by utilizing evolutionary information. It is essential to continually assess the risk of emerging SARS-CoV-2 variants in real time through a combination of computational biology, evolutionary biology, and experimental virology.

## Methods

### Ethics statement

All experiments with hamsters were performed in accordance with the Science Council of Japan's Guidelines for the Proper Conduct of Animal Experiments. The protocols were approved by the Institutional Animal

Care and Use Committee of National University Corporation Hokkaido University (approval ID: 20-0123 and 20-0060). All protocols involving specimens from human subjects recruited at Interpark Kuramochi Clinic was reviewed and approved by the Institutional Review Board of Interpark Kuramochi Clinic (approval ID: G2021-004). All human subjects provided written informed consent. All protocols for the use of human specimens were reviewed and approved by the Institutional Review Boards of The Institute of Medical Science, The University of Tokyo (approval IDs: 2021-1-0416 and 2021-18-0617) and University of Miyazaki (approval ID: O-1021).

## Human serum collection

Convalescent sera were collected from fully vaccinated individuals who had been infected with BA.2 (9 2-dose vaccinated and 5 3-dose vaccinated; 11–61 days after testing. $n = 14$ in total; average age: 47 years, range: 24–84 years, 64% male) (Fig. 3a), and fully vaccinated individuals who had been infected with BA.5 (2 2-dose vaccinated, 17 3-dose vaccinated and 1 4-dose vaccinated; 10–23 days after testing. $n = 20$ in total; average age: 51 years, range: 25–73 years, 45% male) (Fig. 3b). 4-dose vaccine sera from individuals who had been vaccinated with monovalent vaccine (19 donors; average age: 41 years, range: 28–56 years, 42% male) (Fig. 3e), BA.1 bivalent vaccine (22 donors; average age: 55 years, range: 30–73 years, 36% male) (Fig. 3f), and BA.5 bivalent vaccine (21 donors; average age: 51 years, range: 27–86 years, 48% male) (Fig. 3g). The SARS-CoV-2 variants were identified as previously described[2,17,26]. Sera were inactivated at 56 °C for 30 min and stored at −80°C until use. The details of the convalescent sera are summarized in Supplementary Table 3.

## Cell culture

HEK293T cells (a human embryonic kidney cell line; ATCC, CRL-3216), HEK293 cells (a human embryonic kidney cell line; ATCC, CRL-1573) and HOS-ACE2/TMPRSS2 cells (HOS cells stably expressing human ACE2 and TMPRSS2)[41,42] were maintained in DMEM (high glucose) (Sigma-Aldrich, Cat# 6429-500 ML) containing 10% fetal bovine serum (FBS, Sigma-Aldrich Cat# 172012-500 ML) and 1% penicillin–streptomycin (PS) (Sigma-Aldrich, Cat# P4333-100ML). HEK293-ACE2 cells (HEK293 cells stably expressing human ACE2)[22] were maintained in DMEM (high glucose) containing 10% FBS, 1 μg/ml puromycin (InvivoGen, Cat# ant-pr-1) and 1% PS. HEK293-ACE2/TMPRSS2 cells (HEK293 cells stably expressing human ACE2 and TMPRSS2)[22] were maintained in DMEM (high glucose) containing 10% FBS, 1 μg/ml puromycin, 200 μg/ml hygromycin (Nacalai Tesque, Cat# 09287-84) and 1% PS. 293 S GnTI(-) cells (HEK293S cells lacking N-acetylglucosaminyltransferase)[43] were maintained in DMEM (Nacalai tesque, #08458-16) containing 2% FBS without PS. Vero cells [an African green monkey (*Chlorocebus sabaeus*) kidney cell line; JCRB Cell Bank, JCRB0111] were maintained in Eagle's minimum essential medium (EMEM) (Sigma-Aldrich, Cat# M4655-500ML) containing 10% FBS and 1% PS. VeroE6/TMPRSS2 cells (VeroE6 cells stably expressing human TMPRSS2; JCRB Cell Bank, JCRB1819)[44] were maintained in DMEM (low glucose) (Wako, Cat# 041-29775) containing 10% FBS, G418 (1 mg/ml; Nacalai Tesque, Cat# G8168-10ML) and 1% PS. Calu-3 cells (ATCC, HTB-55) were maintained in Eagle's minimum essential medium (EMEM) (Sigma-Aldrich, Cat# M4655-500ML) containing 10% FBS and 1% PS. Calu-3/DSP$_{1-7}$ cells (Calu-3 cells stably expressing DSP$_{1-7}$)[45] were maintained in EMEM (Wako, Cat# 056-08385) containing 20% FBS and 1% PS. Human airway and lung epithelial cells derived from human induced pluripotent stem cells (iPSCs) were manufactured according to established protocols as described below (see "Preparation of human airway and lung epithelial cells from human iPSCs" section) and provided by HiLung Inc. AO-ALI model was generated according to established protocols as described below (see "AO-ALI model" section).

## Viral genome sequencing

Viral genome sequencing was performed as previously described[17]. Briefly, the virus sequences were verified by viral RNA-sequencing analysis. Viral RNA was extracted using a QIAamp viral RNA mini kit (Qiagen, Cat# 52906). The sequencing library employed for total RNA sequencing was prepared using the NEBNext Ultra RNA Library Prep Kit for Illumina (New England Biolabs, Cat# E7530). Paired-end 76-bp sequencing was performed using a MiSeq system (Illumina) with MiSeq reagent kit v3 (Illumina, Cat# MS-102-3001). Sequencing reads were trimmed using fastp v0.21.0[46] and subsequently mapped to the viral genome sequences of a lineage B isolate (strain Wuhan-Hu-1; GenBank accession number: NC_045512.2)[44] using BWA-MEM v0.7.17[47]. Variant calling, filtering, and annotation were performed using SAMtools v1.9[48] and snpEff v5.0e[49].

## Phylogenetic reconstruction

A total of 5,345,749 SARS-CoV-2 genome sequences labeled as 'Omicron' and their corresponding metadata were retrieved from the GISAID database on October 3, 2022 (https://www.gisaid.org/)[50]. The dataset was then filtered based on the following criteria: (i) only 'original passage' sequences, (ii) collection date in 2022, (iii) host labeled as 'Human', (iv) sequence length above 28,000 base pairs and (v) proportion of ambiguous bases below 2%. This filtering reduced the dataset to a total of 3,840,308 sequences. To ensure that PANGO lineage definitions in our dataset's metadata included the latest circulating lineages, the GISAID metadata were downloaded again on October 15, 2022, and PANGO lineages of our sequences were updated accordingly.

To construct an ML tree of Omicron lineages (Fig. 1a), we randomly sampled 100 sequences from BA.1, BA.2, BA.4, and BA.5 and 20 sequences from BA.2.75 and BQ.1.1. In addition, an outgroup sequence, EPI_ISL_466615, representing the oldest isolate of B.1.1 obtained in the UK was added to the dataset. The viral genome sequences were mapped to the reference sequence of Wuhan-Hu-1 (GenBank accession number: NC_045512.2) using Minimap2 v2.17[51] and subsequently converted to a multiple sequence alignment according to the GISAID phylogenetic analysis pipeline (https://github.com/roblanf/sarscov2phylo). The alignment sites corresponding to the 1–265 and 29674–29903 positions in the reference genome were masked (i.e., converted to NNN). Alignment sites at which >50% of sequences contained a gap or undetermined/ambiguous nucleotide were trimmed using trimAl v1.2[52]. Phylogenetic tree construction was performed via a three-step protocol: i) the first tree was constructed; ii) tips with longer external branches (Z score > 4) were removed from the dataset; iii) and the final tree was constructed. Tree reconstruction was performed by RAxML v8.2.12[53] under the GTRCAT substitution model. The node support value was calculated by 100 times bootstrap analysis.

A separate phylogenetic tree was reconstructed for each Omicron lineage (BA.1, BA.2, BA.4 and BA.5) including all their descendant sublineages (Fig. 1c). To remove redundant sequences and reduce the volume of data for each reconstruction, a representative subsampling approach was used. 3000 sequences from each Omicron lineage that had no substitutions at the convergent sites in S: 346, 444, 452, 460 and 486 for BA.1 and BA.2 or no substitutions in sites 346, 444 and 460 for BA.4 and BA.5 were randomly sampled from each dataset, weighting the sampling by the frequency of each PANGO lineage in the dataset. In this way, we included a sample of background sequences with no 'additional' substitutions in the sites of interest with PANGO lineage frequencies representative of the full dataset. It was also ensured that the selected sequences had no ambiguous bases in the S gene (checked between sequence positions 21,000 to 26,000) to avoid ambiguous residues in the sites of interest. Recombinant PANGO lineages were excluded from the analysis.

After collecting the subsampled set of background sequences for each lineage, a maximum of 30 randomly selected sequences of each PANGO sublineage with at least one additional substitution at the convergent sites were added to the dataset. This subsampling approach aimed to capture sequences of all sublineages that have acquired additional mutations at the convergent sites, while maintaining a large set of background lineages that reflects circulating lineage distribution. One SARS-CoV-2 sequence from the sister lineage of each set with a recent collection date was also added to each dataset to be used as an outgroup of the phylogeny [for the BA.1 tree, EPI_ISL_15170885 (BA.2); for the BA.2 tree, EPI_ISL_15148193 (BA.1); for the BA.4 tree, EPI_ISL_15192101 (BA.5); and for BA.5 the tree, EPI_ISL_15174939 (BA.4)].

Each lineage sequence dataset was aligned using the 'global_profile_alignment.sh' from the SARS-CoV-2 global phylogeny pipeline[54], utilizing MAFFT[55]. Phylogenies were reconstructed using iqtree2 (v2.1.3)[56] under a GTR + I + F + G4 model with 1000 ultrafast bootstrap replicates to determine node support. Trees were manually rerooted on the branch leading to the outgroup sequence and time-calibrated with TreeTime[57] (with the '−keep-root' option to preserve the outgroup rooting). Branches leading to tips with dates not matching the root-to-tip regression model were removed from the phylogeny using the ete3 python package[58]. The final trees for BA.1, BA.2, BA.4 and BA.5 contain 3901, 5343, 3328, and 5197 sequences, respectively.

### Ancestral node reconstruction of site substitutions

To infer the branches where substitution events occurred at the five convergent sites (positioned at 346, 444, 452, 460, and 486) in the trees of Omicron lineages, we reconstructed the ancestral state of the substitution profile at the convergent sites in each node using a parsimony method, implemented by the phangorn package (https://github.com/KlausVigo/phangorn). Internal nodes with substitution probabilities above or equal to 0.5 were annotated as having the substitution. Branches where substitutions took place for each site were denoted as branches connecting an ancestral internal node with no substitution to an internal node that has a substitution. Additionally, 70% of tips descending from that internal node were also required to have the substitution and at least 3 tips needed to be descended from the node, to avoid picking up branches with low support or clades that reverted back to the original residue. The analysis was performed on R v4.1.2 (https://www.r-project.org/).

Regarding the threshold of minimum probability to define that a substitution exists at each internal node, we performed a sensitivity analysis and confirmed that the number of detected substitution events is robust to the setting of the threshold (Supplementary Fig. 1b).

### Modeling the relationship between viral epidemic dynamics and S substitutions

Motivated by the model established by Obermeyer et al.[40], we developed a method to model the relationship between viral epidemic dynamics and S substitutions. This model can simultaneously estimate (i) the effect of each S substitution on $R_e$ and (ii) the relative $R_e$ of a viral group represented by each S haplotype. The key concept of the model used in this study is the same as the one in Obermeyer et al.[40]. However, our method is independent of the predefined viral classification such as PANGO lineage but based on the viral classification according to the profile of S substitutions. Therefore, our method can link the effect of S substitutions to viral epidemic dynamics in a more direct manner. Also, in our method, a Markov Chain Monte Carlo (MCMC) method is used for parameter estimation instead of variational inference, an approximation method.

The data used in this analysis were downloaded from the GISAID database (https://www.gisaid.org/) on November 7, 2022. For quality control, we excluded the data of viral sequences with the following features from the analysis: (i) a lack of collection date information; (ii) sampling in animals other than humans; (iii) >1% undetermined nucleotide characters; or (iv) sampling by quarantine. Furthermore, in this analysis, we analyzed viral sequences of the Omicron lineages collected in the UK from March 1, 2022, to October 15, 2022.

We selected S substitutions (including insertions and deletions) to be analyzed and classified Omicron sequences into S haplotypes according to the profile of the selected S substitutions: We analyzed S substitutions observed in ≥200 sequences in the dataset we used. We excluded S substitutions commonly (≥90%) detected in sequences analyzed. According to the criteria above, 123 S substitutions were retrieved. Subsequently, we classified the sequences according to the profile of S substitutions above (referred to as S haplotype). We excluded S haplotypes with ≤30 sequences from the downstream analyses. According to the criterion above, 254 S haplotypes, composed of 375,121 sequences, were retrieved. The substitution profile was represented as a matrix, where the rows and columns depict S haplotypes and S substitutions, respectively. An element in the matrix represents the status [presence (1) or absence (0)] of one S substitution in one S haplotype. Next, we identified a group of highly co-occurring substitutions (i.e., a pair of substitutions with >0.9 Pearson's correlation in the substitution profile matrix) and clustered these substitutions as a substitution cluster (Supplementary Fig. 1d). For example, the L452R:F486V cluster represents the L452R and F486V substitutions. For one substitution cluster, the mean value of the substitution statuses (0 or 1) of the members of substitutions was calculated for each S haplotype, and the mean value was used as the substitution status of the substitution cluster. For example, if one S haplotype has L452R but not F486V, the substitution status of the L452R:F486V cluster of the haplotype was set at 0.5. Consequently, our dataset included the profile of 107 S substitutions/substitution clusters for 254 S haplotypes. Next, to set the major S haplotype of BA.2 as the reference S haplotype (or lineage) in the statistical model described below, we transformed the S substitution profile matrix by subtracting the substitution profile of the major S haplotype of BA.2 from those for all S haplotypes. Consequently, elements in the transformed S substitution profile matrix were converted to −1, 0, or 1: The zero value means that the status of a substitution in one haplotype is the same as that in the reference haplotype. The one value means that a substitution is present in one haplotype but not in the reference haplotype. The minus one value means that a substitution is absent in one haplotype but present in the reference haplotype. As a consequence of the transformation, the relative $R_e$ value for the reference haplotype was set at 1 in the parameter estimation in the statistical model described below. Finally, the number of viral sequences belonging to each S haplotype collected on each day was counted, and the count matrix was constructed as an input for the statistical model described below.

We assigned one major lineage classification (i.e., BA.1 BA.2, BA.4, BA.5, and BA.2.75) to each S haplotype: We examined the major lineage classification of respective viral sequences belonging to one S haplotype, and the classification of the S haplotype was determined according to the majority vote system.

We constructed a Bayesian hierarchal model, which represents the epidemic dynamics of each S haplotype according to growth rate parameters for each S haplotype, which is represented by a linear combination of the effect of S substitutions. Arrays in the model index over one or more indices: $L = 254$ viral lineages (i.e., S haplotypes) l; $S = 107$ substitutions/substitution clusters s; and $T = 229$ days t. The model is:

$$\sigma_1 \sim Student\_t^+(5,0,10)$$

$$f_m \sim Laplace(0,10)$$

$$\beta_l \sim \text{Student } t\left(5, \sum_m f_m X_{lm}, \sigma_1\right)$$

$$y_{\cdot t} \sim \text{Multinomial}\left(\sum_l y_{lt}, softmax\left(\alpha_{\cdot} + \beta_{\cdot}t\right)\right)$$

The count of viral lineage l at time t, $y_{lt}$, is modeled as a hierarchal Multinomial logistic regression with intercept $\alpha_l$ and slope $\beta_l$ parameters for lineage l. The slope (or viral lineage growth) parameter $\beta_l$ is generated from Student's t distribution with five degrees of freedom, the mean value represented by $f_m X_{lm}$, and standard deviation, $\sigma_1$. $f_m X_{lm}$ denotes the linear combination of the effect of each substitution, where $f_m$ and $X_{lm}$ are the effect of substitution m and the profile of substitution m in lineage l (i.e., the substitution profile matrix constructed in the above paragraph), respectively. As a prior of $f_m$, the Laplace distribution with the mean 0 and the standard deviation 10 was set. In other words, we estimated the parameter $f_m$ in the framework of Bayesian least absolute shrinkage and selection operator (LASSO). As a prior of $\sigma_1$, a half Student's t distribution with the mean 0 and the standard deviation 10 was set. For the other parameters, non-informative priors were set.

The relative $R_e$ of each viral lineage, $r_l$, was calculated according to the slope parameter $\beta_l$ as

$$r_l = \exp(\gamma \beta_l)$$

where $\gamma$ is the average viral generation time (2.1 days) (http://sonorouschocolate.com/covid19/index.php?title=Estimating_Generation_Time_Of_Omicron). Similarly, the effect size of substitution m on the relative $R_e$, $F_l$, was calculated according to the coefficient $f_l$ as:

$$F_l = \exp(\gamma f_l)$$

Parameter estimation was performed via the MCMC approach implemented in CmdStan v2.30.1 (https://mc-stan.org) with CmdStanr v0.5.3 (https://mc-stan.org/cmdstanr/). Four independent MCMC chains were run with 500 and 2000 steps in the warmup and sampling iterations, respectively. We confirmed that all estimated parameters showed <1.01 R-hat convergence diagnostic values and >200 effective sampling size values, indicating that the MCMC runs were successfully convergent. The above analyses were performed in R v4.2.1 (https://www.r-project.org/). Information on the estimated effect size of each substitution or substitution cluster on relative $R_e$ and relative $R_e$ for each S haplotype are summarized in Supplementary Data 1, 2.

Since our model simply represents the viral lineage growth parameter ($\beta_l$) as the linear combination of the effects of S substitutions, the model can predict the total effect of a set of substations on relative $R_e$. Using this property of the model, we predicted (i) the relative $R_e$ of S haplotypes in countries other than UK (Supplementary Fig. 2), (ii) the total effect of substitutions at the convergent sites (Fig. 2f), and iii) the ancestral relative viral fitness for each internal node in the BA.5 tree (Fig. 2g).

To evaluate the generalization ability of our model, we predicted the relative $R_e$ of S haplotypes in 20 countries with a higher amount sequencing data using the model trained on the UK's data. First, the genome surveillance data downloaded from GISAID on November 7, 2022, were filtered according to the criteria same to the ones applied for the UK's data described above. Second, viral sequences were classified into S haplotypes according to the profile of 107 S substitutions/substitution clusters, used for the haplotype classification in the UK's dataset. Third, in each country, S haplotypes with ≥30 sequences were used for the downstream analyses. Fourth, we estimated the

relative $R_e$ for each S haplotype in each country by a simple multi-nomial logistic model described elsewhere. The S haplotype corresponding to the major S haplotype of BA.2 in the UK was selected as the reference, and $R_e$ for the reference haplotype was set at 1. Finally, we predicted the expected value of relative $R_e$ of each S haplotype ($l$) ($r_{l_{pred}}$) as the following formula $E(r_{l_{pred}}) = E(\exp(\gamma \sum_m f_m X_{lm}))$ using parameters in the model trained on the UK' data. The consistency of the relative $R_e$ estimated by a simple multinomial logistic model in each country and $R_e$ predicted by the model trained on the UK's data was evaluated by adjusted $R^2$ value.

To evaluate how much proportion of the variance of relative $R_e$ in Omicron lineage can be explained by the profile of substitutions at the five convergent sites, we calculated the total effect of substitutions at the convergent sites in the UK's data. For the calculation of the total effect, the formula described above was basically used, but the profile of only convergent substitutions was considered.

To predict the ancestral relative viral fitness for each internal node in the BA.5 tree, we first reconstructed the ancestral state of the S substitution profile in each node of the tree using a parsimony method, implemented by the phangorn package. Subsequently, we predicted the relative viral fitness for each node according to the reconstructed ancestral mutation profile for the node suing the formula described above. The above analyses were performed in R v4.2.1 (https://www.r-project.org/).

## Plasmid construction

Plasmids expressing the codon-optimized SARS-CoV-2 S proteins of B.1.1 (the parental D614G-bearing variant), BA.2 and BA.5, and BA.2.75 were prepared in our previous studies[2,17,22,26]. Plasmids expressing the codon-optimized S proteins of BQ.1.1, BA.5 S-based derivatives and BA.2 S-based derivatives were generated by site-directed overlap extension PCR using the primers listed in Supplementary Table 4. The resulting PCR fragment was digested with KpnI (New England Biolabs, Cat# R0142S) and NotI (New England Biolabs, Cat# R1089S) and inserted into the corresponding site of the pCAGGS vector[59]. Nucleotide sequences were determined by DNA sequencing services (Eurofins), and the sequence data were analyzed by Sequencher v5.1 software (Gene Codes Corporation). Plasmids for yeast surface display were constructed by restriction enzyme-free cloning by incorporation of RBD genes ["construct 3" in ref. 24, covering residues 330–528] into the pJYDC1 plasmid (Addgene, Cat# 162458). The primers are listed in Supplementary Table 4. The non-mutated RBD genes (BA.2, BA.5, and BQ.1) were purchased from Twist Biosciences.

## Neutralization assay

Pseudoviruses were prepared as previously described[2,17,22,25,26,31,32,45,60–64]. Briefly, lentivirus (HIV-1)-based, luciferase-expressing reporter viruses were pseudotyped with SARS-CoV-2 S proteins. HEK293T cells (1,000,000 cells) were cotransfected with 1 µg psPAX2-IN/HiBiT[41], 1 µg pWPI-Luc2[41], and 500 ng plasmids expressing parental S or its derivatives using PEI Max (Polysciences, Cat# 24765-1) according to the manufacturer's protocol. Two days posttransfection, the culture supernatants were harvested and centrifuged. The pseudoviruses were stored at −80°C until use.

The neutralization assay (Fig. 3) was prepared as previously described[2,17,25,26,31,45,60–64]. Briefly, the SARS-CoV-2 S pseudoviruses (counting ~20,000 relative light units) were incubated with serially diluted (120-fold to 87,480-fold dilution at the final concentration) heat-inactivated sera at 37°C for 1 h. Pseudoviruses without sera were included as controls. Then, a 40 µl mixture of pseudovirus and serum/antibody was added to HOS-ACE2/TMPRSS2 cells (10,000 cells/50 µl) in a 96-well white plate. At 2 d.p.i., the infected cells were lysed with a One-Glo luciferase assay system (Promega, Cat# E6130), a Bright-Glo luciferase assay system (Promega, Cat# E2650), or a britelite plus Reporter Gene Assay System (PerkinElmer, Cat# 6066769), and the

luminescent signal was measured using a GloMax explorer multimode microplate reader 3500 (Promega) or CentroXS3 LB960 (Berthhold Technologies). The assay of each serum sample was performed in triplicate, and the 50% neutralization titer ($NT_{50}$) was calculated using Prism 9 software v9.1.1 (GraphPad Software).

### SARS-CoV-2 preparation and titration

The working virus stocks of SARS-CoV-2 were prepared and titrated as previously described[2,17,22,26,31,32,63–65]. In this study, clinical isolates of B.1.1 (strain TKYE610670; GISAID ID: EPI_ISL_479681)[31], Delta (B.1.617.2, strain TKYTK1734; GISAID ID: EPI_ISL_2378732)[32], BA.2 (strain TY40-385; GISAID ID: EPI_ISL_9595859)[17], BA.5 (strain TKYS14631; GISAID ID: EPI_ISL_12812500)[2,35], and BQ.1.1 (strain TY41-796-P1; GISAID ID: EPI_ISL_15579783) were used. In brief, 20 µl of the seed virus was inoculated into VeroE6/TMPRSS2 cells (5,000,000 cells in a T-75 flask). One hour post-infection (h.p.i.), the culture medium was replaced with DMEM (low glucose) (Wako, Cat# 041-29775) containing 2% FBS and 1% PS. At 3 d.p.i., the culture medium was harvested and centrifuged, and the supernatants were collected as the working virus stock.

The titer of the prepared working virus was measured as $TCID_{50}$. Briefly, one day before infection, VeroE6/TMPRSS2 cells (10,000 cells) were seeded into a 96-well plate. Serially diluted virus stocks were inoculated into the cells and incubated at 37°C for 4 days. The cells were observed under a microscope to judge the CPE appearance. The value of $TCID_{50}$/ml was calculated with the Reed–Muench method[66].

For verification of the sequences of SARS-CoV-2 working viruses, viral RNA was extracted from the working viruses using a QIAamp viral RNA mini kit (Qiagen, Cat# 52906) and viral genome sequences were analyzed as described above (see "Viral genome sequencing" section). Information on the unexpected substitutions detected is summarized in Supplementary Table 5, and the raw data are deposited in the GitHub repository (https://github.com/TheSatoLab/BQ.1).

### Yeast surface display

Yeast surface display (Fig. 4a) was performed as previously described[2,23,24]. Briefly, the *S. cerevisiae* EBY100 yeasts were transformed with RBD expression plasmid and grown (220 rpm, 30°C, SD-CAA media). The expression media 1/9[67] was inoculated to starting $OD_{600}$ 0.7–1 by overnight grown culture and cultivated for 24 h at 20°C. The expression media was supplemented with 10 nM DMSO solubilized bilirubin (Sigma-Aldrich, Cat# 14370-1 G) for activation of eUnaG2 fluorescence (excitation at 498 nm, emission at 527 nm).

Yeast cells were washed in ice-cold PBSB buffer (PBS with 1 mg/ml BSA), liquated (100 µl), transferred in an analysis solution and incubated for 8 h. The analysis solutions consisted of a series of CF®640 R succinimidyl ester labeled (Biotium, Cat# 92108) ACE2 peptidase domain (residues 18–740) concentrations, PBSB buffer and 1 nM bilirubin. Incubated samples were washed twice with PBSB buffer and transferred into a 96-well plate (Thermo Fisher Scientific, Cat# 268200) for automated data acquisition by a CytoFLEX S Flow Cytometer (Beckman Coulter, USA, Cat#. N0-V4-B2-Y4). The gating and analysis strategies were described previously[24]. The titration curves were fitted with nonlinear least-squares regression using Python v3.7 and two additional parameters to describe the titration curve[24].

### Pseudovirus infection

Pseudovirus infection (Fig. 4b) was performed as previously described[2,17,22,25,26,31,32,45,60–64]. Briefly, the amount of pseudoviruses prepared was quantified by the HiBiT assay using a Nano Glo HiBiT lytic detection system (Promega, Cat# N3040) as previously described[41,68]. For measurement of pseudovirus infectivity, the same amount of pseudoviruses (normalized to the HiBiT value, which indicates the amount of HIV-1 p24 antigen) was inoculated into HOS-ACE2/TMPRSS2 cells, HEK293-ACE2 cells or HEK293-ACE2/TMPRSS2 cells and viral infectivity was measured as described above (see "Neutralization

assay" section). For analysis of the effect of TMPRSS2 on pseudovirus infectivity (Fig. 4c), the fold change of the values of HEK293-ACE2/TMPRSS2 to HEK293-ACE2 was calculated.

### Protein expression and purification of BQ.1.1 S RBD and human ACE2

The BQ.1.1 S RBD and human ACE2 were prepared as previously described[69]. Briefly, the expression plasmids encoding the BQ.1.1 S RBD (residues 322-536) or human ACE2 (residues 19-617) were transfected into 293 S GnTI (-) cells. The proteins in the culture supernatant were purified with cOmplete His-Tag Purification Resin (Roche) affinity column, followed by Superdex 75 Increase 10/300 size-exclusion chromatography (Cytiva) with a running buffer containing of 0.1 M Imidazole pH8.0, 150 mM NaCl.

### Crystallization and data collection

The sitting-drop method was used to obtain the BQ.1.1 S RBD-human ACE2 complex crystals. In detail, purified complex proteins were concentrated to 20 mg/ml. Then, 0.4 µl protein was mixed with 0.4 µl reservoir solution. The resulting solution was sealed and equilibrated against 50 µl reservoir solution at 293 K. Crystals of the BQ.1.1 S RBD-human ACE2 complex were grown in 0.1 M sodium acetate (pH 4-4.5), 0.1–0.2 M ammonium acetate, 13.2–17.4% polyethylene glycol 4000. Then, crystals were soaked briefly in cryoprotectant mixture of precipitant and 80% glycerol in a ratio of 6.5:3.5 ratio and flash frozen in liquid nitrogen. X-ray diffraction data were collected from beamline BL32XU at Spring-8 (Hyogo, Japan). All diffraction data were indexed, integrated, scaled, and merged using ZOO system[70] including KAMO[71] and XDS[72].

### Structure determination and refinement

The crystal structure of the BQ.1.1 RBD-hACE2 complex (Fig. 4d–f) was determined by the molecular replacement method with Phaser[73], using previously reported SARS-CoV-2 BA.4/5 variant RBD-human ACE2 complex structure (PDB: 7XWA)[17] as search models. The initial protein models were rebuilt using ModelCraft[74] and fitted manually using Coot[75]. The structure was then refined using phenix.refine[76]. The data collection and refinement statistics are summarized in Supplementary Table 2. All structure figures were generated by PyMOL (https://pymol.org/2/).

### SARS-CoV-2 S-based fusion assay

A SARS-CoV-2 S-based fusion assay (Fig. 5a, b and Supplementary Fig. 3e) was performed as previously described[2,17,22,26,31,63]. Briefly, on day 1, effector cells (i.e., S-expressing cells) and target cells (Calu-3/DSP$_{1-7}$ cells) were prepared at a density of 0.6–0.8 × 10⁶ cells in a 6-well plate. On day 2, for the preparation of effector cells, HEK293 cells were cotransfected with the S expression plasmids (400 ng) and pDSP$_{8-11}$ (ref. 77) (400 ng) using TransIT-LT1 (Takara, Cat# MIR2300). On day 3 (24 h posttransfection), 16,000 effector cells were detached and reseeded into a 96-well black plate (PerkinElmer, Cat# 6005225), and target cells were reseeded at a density of 1,000,000 cells/2 ml/well in 6-well plates. On day 4 (48 h posttransfection), target cells were incubated with EnduRen live cell substrate (Promega, Cat# E6481) for 3 h and then detached, and 32,000 target cells were added to a 96-well plate with effector cells. *Renilla* luciferase activity was measured at the indicated time points using Centro XS3 LB960 (Berthhold Technologies). For measurement of the surface expression level of the S protein, effector cells were stained with rabbit anti-SARS-CoV-2 S S1/S2 polyclonal antibody (Thermo Fisher Scientific, Cat# PA5-112048, 1:100). Normal rabbit IgG (Southern Biotech, Cat# 0111-01, 1:100) was used as a negative control, and APC-conjugated goat anti-rabbit IgG polyclonal antibody (Jackson ImmunoResearch, Cat# 111-136-144, 1:50) was used as a secondary antibody. The surface expression level of S proteins (Supplementary Fig. 3e) was measured using a FACS Canto II (BD

Biosciences) and the data were analyzed using FlowJo software v10.7.1 (BD Biosciences). Gating strategy for flow cytometry is shown in Supplementary Fig. 5. For calculation of fusion activity, *Renilla* luciferase activity was normalized to the mean fluorescence intensity (MFI) of surface S proteins. The normalized value (i.e., *Renilla* luciferase activity per the surface S MFI) is shown as fusion activity.

## AO-ALI model

An airway organoid (AO) model was generated according to our previous report[2,78]. Briefly, normal human bronchial epithelial cells (NHBEs, Cat# CC-2540, Lonza) were used to generate AOs. NHBEs were suspended in 10 mg/ml cold Matrigel growth factor reduced basement membrane matrix (Corning, Cat# 354230). Fifty microliters of cell suspension were solidified on prewarmed cell culture-treated multiple dishes (24-well plates; Thermo Fisher Scientific, Cat# 142475) at 37 °C for 10 min, and then, 500 μl of expansion medium was added to each well. AOs were cultured with AO expansion medium for 10 days. For maturation of the AOs, expanded AOs were cultured with AO differentiation medium for 5 days.

The AO-ALI model (Fig. 5f) was generated according to our previous report[2,78]. For generation of AO-ALI, expanding AOs were dissociated into single cells, and then were seeded into Transwell inserts (Corning, Cat# 3413) in a 24-well plate. AO-ALI was cultured with AO differentiation medium for 5 days to promote their maturation. AO-ALI was infected with SARS-CoV-2 from the apical side.

## Preparation of human airway and alveolar epithelial cells from human iPSCs

The air-liquid interface culture of airway and alveolar epithelial cells (Fig. 5g, h) was differentiated from human iPSC-derived lung progenitor cells as previously described[2,17,35,79–81]. Briefly, alveolar progenitor cells were induced stepwise from human iPSCs according to a 21-day and 4-step protocol[79]. At day 21, alveolar progenitor cells were isolated with the specific surface antigen carboxypeptidase M and seeded onto the upper chamber of a 24-well Cell Culture Insert (Falcon, #353104), followed by 28-day and 7-day differentiation of airway and alveolar epithelial cells, respectively. Alveolar differentiation medium with dexamethasone (Sigma-Aldrich, Cat# D4902), KGF (PeproTech, Cat# 100-19), 8-Br-cAMP (Biolog, Cat# B007), 3-isobutyl 1-methylxanthine (IBMX) (Fujifilm Wako, Cat# 095-03413), CHIR99021 (Axon Medchem, Cat# 1386), and SB431542 (Fujifilm Wako, Cat# 198-16543) was used for the induction of alveolar epithelial cells. PneumaCult ALI (STEMCELL Technologies, Cat# ST-05001) with heparin (Nacalai Tesque, Cat# 17513-96) and Y-27632 (LC Laboratories, Cat# Y-5301) hydrocortisone (Sigma-Aldrich, Cat# H0135) was used for induction of airway epithelial cells.

## Airway-on-a-chips

Airway-on-a-chips (Fig. 5i) were prepared as previously described[2,34,35]. Human lung microvascular endothelial cells (HMVEC-L) were obtained from Lonza (Cat# CC-2527) and cultured with EGM-2-MV medium (Lonza, Cat# CC-3202). For preparation of the airway-on-a-chip, first, the bottom channel of a polydimethylsiloxane (PDMS) device was precoated with fibronectin (3 μg/ml, Sigma-Aldrich, Cat# F1141). The microfluidic device was generated according to our previous report[82]. HMVEC-L cells were suspended at 5,000,000 cells/ml in EGM2-MV medium. Then, 10 μl of suspension medium was injected into the fibronectin-coated bottom channel of the PDMS device. Then, the PDMS device was turned upside down and incubated. After 1 h, the device was turned over, and the EGM-2-MV medium was added into the bottom channel. After 4 days, AOs were dissociated and seeded into the top channel. AOs were generated according to our previous report[78]. AOs were dissociated into single cells and then suspended at 5,000,000 cells/ml in the AO differentiation medium. Ten microliter suspension medium was injected into the top channel. After 1 h, the AO

differentiation medium was added to the top channel. In the infection experiments (Fig. 5i), the AO differentiation medium containing either BA.2, BA.5, BQ.1.1 or Delta isolate (500 TCID$_{50}$) was inoculated into the top channel. At 2 h.p.i., the top and bottom channels were washed and cultured with AO differentiation and EGM2-MV medium, respectively. The culture supernatants were collected, and viral RNA was quantified using RT–qPCR (see "RT–qPCR" section above).

## Microfluidic device

A microfluidic device was generated according to our previous report[2,82]. Briefly, the microfluidic device consisted of two layers of microchannels separated by a semipermeable membrane. The microchannel layers were fabricated from PDMS using a soft lithographic method. PDMS prepolymer (Dow Corning, Cat# SYLGARD 184) at a base to curing agent ratio of 10:1 was cast against a mold composed of SU-8 2150 (MicroChem, Cat# SU-8 2150) patterns formed on a silicon wafer. The cross-sectional size of the microchannels was 1 mm in width and 330 μm in height. Access holes were punched through the PDMS using a 6-mm biopsy punch (Kai Corporation, Cat# BP-L60K) to introduce solutions into the microchannels. Two PDMS layers were bonded to a PET membrane containing 3.0-μm pores (Falcon, Cat# 353091) using a thin layer of liquid PDMS prepolymer as the mortar. PDMS prepolymer was spin-coated (4000 rpm for 60 s) onto a glass slide. Subsequently, both the top and bottom channel layers were placed on the glass slide to transfer the thin layer of PDMS prepolymer onto the embossed PDMS surfaces. The membrane was then placed onto the bottom layer and sandwiched with the top layer. The combined layers were left at room temperature for 1 day to remove air bubbles and then placed in an oven at 60 °C overnight to cure the PDMS glue. The PDMS devices were sterilized by placing them under UV light for 1 h before the cell culture.

## SARS-CoV-2 infection

One day before infection, Vero cells (10,000 cells), VeroE6/TMPRSS2 cells (10,000 cells) and Calu-3 cells (10,000 cells) were seeded into a 96-well plate. SARS-CoV-2 [1,000 TCID$_{50}$ for Vero cells (Fig. 5c); 100 TCID$_{50}$ for VeroE6/TMPRSS2 cells (Fig. 5d) and Calu-3 cells (Fig. 5e)] was inoculated and incubated at 37 °C for 1 h. The infected cells were washed, and 180 μl of culture medium was added. The culture supernatant (10 μl) was harvested at the indicated timepoints and used for RT–qPCR to quantify the viral RNA copy number (see "RT–qPCR" section below). In the infection experiments using human iPSC-derived airway and lung epithelial cells (Fig. 5g, h), working viruses were diluted with Opti-MEM (Thermo Fisher Scientific, Cat# 11058021). The diluted viruses (1000 TCID$_{50}$ in 100 μl) were inoculated onto the apical side of the culture and incubated at 37 °C for 1 h. The inoculated viruses were removed and washed twice with Opti-MEM. For the collection of the viruses, 100 μl Opti-MEM was applied onto the apical side of the culture and incubated at 37 °C for 10 min. The Opti-MEM was collected and used for RT–qPCR to quantify the viral RNA copy number (see "RT–qPCR" section below). The infection experiments using an airway-on-a-chip system (Fig. 5i) were performed as described above (see "Airway-on-a-chips" section).

## RT–qPCR

RT–qPCR was performed as previously described[2,17,22,26,31,32,63–65]. Briefly, 5 μl culture supernatant was mixed with 5 μl of 2 × RNA lysis buffer [2% Triton X-100 (Nacalai Tesque, Cat# 35501-15), 50 mM KCl, 100 mM Tris-HCl (pH 7.4), 40% glycerol, 0.8 U/μl recombinant RNase inhibitor (Takara, Cat# 2313B)] and incubated at room temperature for 10 min. RNase-free water (90 μl) was added, and the diluted sample (2.5 μl) was used as the template for real-time RT-PCR performed according to the manufacturer's protocol using One Step TB Green PrimeScript PLUS RT-PCR kit (Takara, Cat# RR096A) and the following primers: Forward *N*, 5′-AGC CTC TTC TCG TTC CTC ATC AC-3′; and Reverse *N*, 5′-CCG

CCA TTG CCA GCC ATT C-3′. The viral RNA copy number was standardized with a SARS-CoV-2 direct detection RT-qPCR kit (Takara, Cat# RC300A). Fluorescent signals were acquired using a QuantStudio 1 Real-Time PCR system (Thermo Fisher Scientific), QuantStudio 3 Real-Time PCR system (Thermo Fisher Scientific), QuantStudio 5 Real-Time PCR system (Thermo Fisher Scientific), StepOne Plus Real-Time PCR system (Thermo Fisher Scientific), CFX Connect Real-Time PCR Detection system (Bio-Rad), Eco Real-Time PCR System (Illumina), qTOWER3 G Real-Time System (Analytik Jena) Thermal Cycler Dice Real Time System III (Takara) or 7500 Real-Time PCR System (Thermo Fisher Scientific).

### Animal experiments

Animal experiments (Fig. 6 and Supplementary Fig. 4) were performed as previously described[2,17,26,31,32,35]. Syrian hamsters (male, 4 weeks old) were purchased from Japan SLC Inc. (Shizuoka, Japan). For the virus infection experiments, hamsters were anesthetized by intramuscular injection of a mixture of 0.15 mg/kg medetomidine hydrochloride (Domitor®, Nippon Zenyaku Kogyo), 2.0 mg/kg midazolam (Dormicum®, Fujifilm Wako, Cat# 135-13791) and 2.5 mg/kg butorphanol (Vetorphale®, Meiji Seika Pharma) or 0.15 mg/kg medetomidine hydrochloride, 4.0 mg/kg alphaxaone (Alfaxan®, Jurox) and 2.5 mg/kg butorphanol. BA.5, BQ.1.1 and Delta (10,000 $TCID_{50}$ in 100 μl) or saline (100 μl) was intranasally inoculated under anesthesia. Oral swabs were collected at the indicated timepoints. Body weight was recorded daily by 7 d.p.i. Enhanced pause (Penh), the ratio of time to peak expiratory follow relative to the total expiratory time (Rpef) were measured every day until 7 d.p.i. (see below). Lung tissues were anatomically collected at 2 and 5 d.p.i. The viral RNA load in the oral swabs and respiratory tissues was determined by RT−qPCR. These tissues were also used for IHC and histopathological analyses (see below).

### Lung function test

Lung function tests (Fig. 6a) were routinely performed as previously described[2,17,26,31,35]. The two respiratory parameters (Penh and Rpef) were measured by using a Buxco Small Animal Whole Body Plethysmography system (DSI) according to the manufacturer's instructions. In brief, a hamster was placed in an unrestrained plethysmography chamber and allowed to acclimatize for 30 s. Then, data were acquired over a 2.5-min period by using FinePointe Station and Review software v2.9.2.12849 (DSI).

### Immunohistochemistry

Immunohistochemical (IHC) analysis (Fig. 6c and Supplementary Fig. 4) was performed as previously described[2,17,26,31,35] using an Autostainer Link 48 (Dako). The deparaffinized sections were exposed to EnVision FLEX target retrieval solution high pH (Agilent, Cat# K8004) for 20 min at 97℃ for activation, and a mouse anti-SARS-CoV-2 N monoclonal antibody (clone 1035111, R&D Systems, Cat# MAB10474-SP, 1:400) was used as a primary antibody. The sections were sensitized using EnVision FLEX for 15 min and visualized by peroxidase-based enzymatic reaction with 3,3′-diaminobenzidine tetrahydrochloride (Dako, Cat# DM827) as substrate for 5 min. The N protein positivity was evaluated by certificated pathologists as previously described[2,17,26,31,35]. Images were incorporated as virtual slides by NDP.scan software v3.2.4 (Hamamatsu Photonics). The N-protein positivity was measured as the area using Fiji software v2.2.0 (ImageJ).

### H&E staining

Hematoxylin and eosin (H&E) staining (Fig. 6d) was performed as previously described[2,17,26,31,35]. Briefly, excised animal tissues were fixed with 10% formalin neutral buffer solution and processed for paraffin embedding. The paraffin blocks were sectioned at a thickness of 3 μm and then mounted on MAS-GP-coated glass slides (Matsunami Glass, Cat# S9901). H&E staining was performed according to a standard protocol.

### Histopathological scoring

Histopathological scoring (Fig. 6e) was performed as previously described[2,17,26,31,35]. Pathological features, including (i) bronchitis or bronchiolitis, (ii) hemorrhage with congestive edema, (iii) alveolar damage with epithelial apoptosis and macrophage infiltration, (iv) hyperplasia of type II pneumocytes, and (v) the area of hyperplasia of large type II pneumocytes, were evaluated by certified pathologists, and the degree of these pathological findings was arbitrarily scored using a four-tiered system as 0 (negative), 1 (weak), 2 (moderate), and 3 (severe). The "large type II pneumocytes" are type II pneumocytes with hyperplasia exhibiting more than 10-μm-diameter nuclei. We described "large type II pneumocytes" as one of the notable histopathological features of SARS-CoV-2 infection in our previous studies[2,17,26,31,35]. The total histological score is the sum of these five indices.

### Statistics and reproducibility

Statistical significance was tested using a two-sided Mann–Whitney $U$ test, a two-sided Student's $t$ test, a two-sided Welch's $t$ test, or a two-sided paired $t$-test unless otherwise noted. The tests above were performed using Prism 9 software v9.1.1 (GraphPad Software).

In the time-course experiments (Figs. 5, 6a–b, e), a multiple regression analysis including experimental conditions (i.e., the types of infected viruses) as explanatory variables and timepoints as qualitative control variables was performed to evaluate the difference between experimental conditions thorough all timepoints. The initial time point was removed from the analysis. The $P$ value was calculated by a two-sided Wald test. Subsequently, familywise error rates (FWERs) were calculated by the Holm method. These analyses were performed on R v4.1.2 (https://www.r-project.org/).

Principal component analysis to representing the antigenicity of the S proteins was performed (Fig. 3d). The $NT_{50}$ values for biological replicates were scaled, and subsequently, principal component analysis was performed using the prcomp function on R v4.1.2 (https://www.r-project.org/).

In Fig. 6c, d, and Supplementary Fig. 4, photographs shown are the representative areas of at least two independent experiments by using four hamsters at each timepoint.

### Reporting summary

Further information on research design is available in the Nature Portfolio Reporting Summary linked to this article.

## Data availability

All databases/datasets used in this study are available from the GISAID database (https://www.gisaid.org; EPI_SET_221203cz, EPI_SET_221203ep, EPI_SET_221203qr, EPI_SET_221203se, and EPI_SET_230302mz) and GenBank database (https://www.ncbi.nlm.nih.gov/genbank/). Viral genome sequencing data for working viral stocks are available in the GitHub repository (https://github.com/TheSatoLab/BQ.1). Source data are provided with this paper.

The atomic coordinate for the crystal structure of the BQ.1.1 RBD-human ACE2 complex determined in this study are available in the Protein Data Bank (www.rcsb.org) under accession code 8IF2 Source data are provided with this paper.

## Code availability

The computational codes used in the present study are available in the GitHub repository (https://github.com/TheSatoLab/BQ.1).

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

## Acknowledgements

The authors would like to thank all members belonging to The Genotype to Phenotype Japan (G2P-Japan) Consortium. We thank Dr. Kenzo Tokunaga (National Institute for Infectious Diseases, Japan) and Dr. Jin Gohda (The University of Tokyo, Japan) for providing reagents. We also thank the National Institute for Infectious Diseases, Japan, for providing clinical isolates of BQ.1.1 (strain TY41-796-P1; GISAID ID: EPI_ISL_15579783) and BA.2 (strain TY40-385; GISAID ID: EPI_ISL_9595859). We appreciate the technical assistance to the BL32XU beamline staff at SPring-8 and from The Research Support Center, Research Center for Human Disease Modeling, Kyushu University Graduate School of Medical Sciences. We gratefully acknowledge all data contributors, i.e., the Authors and their Originating laboratories responsible for obtaining the specimens, and their Submitting laboratories for generating the genetic sequence and metadata and sharing via the GISAID Initiative, on which this research is based. The super-computing resource was provided by the Human Genome Center at The University of Tokyo. This study was supported in part by AMED SCARDA Japan Initiative for World-leading Vaccine Research and Development Centers "UTOPIA" (JP223fa627001, to K.S.), AMED SCARDA Program on R&D of new generation vaccine including new modality application (JP223fa727002, to K.S.); AMED SCRADA Kyoto University Immunomonitoring Center (KIC) (JP223fa627009, to T.H.); AMED SCARDA World-leading institutes for vaccine research and development Hokkaido

Synergy Campus (223fa627005h0001, to K.M., and T.F.); AMED Research Program on Emerging and Re-emerging Infectious Diseases (JP21fk0108574, to H.N.; JP21fk0108465, to A.S.; JP22fk0108516, to T.F.; JP21fk0108493, to Takasuke Fukuhara; JP22fk0108617 to T.F.; JP22fk0108146, to K.S.; JP21fk0108494 to G2P-Japan Consortium, K.M., S.T., T.I., T.F., and K.S.; JP21fk0108425, to K.T., A.S., and K.S.; 22fk0108506, to K.T., A.S., and K.S.; JP21fk0108432, to K.T., T.F., and K.S.); AMED Research Program on HIV/AIDS (JP22fk0410033, to A.S.; JP22fk0410047, to A.S.; JP22fk0410055, to T.I.; and JP22fk0410039, to K.S.); AMED CRDF Global Grant (JP22jk0210039 to A.S.); AMED Japan Program for Infectious Diseases Research and Infrastructure (JP22wm0325009, to A.S.; JP22wm0125008 to K.M.); AMED CREST (JP21gm1610005, to K.T.; JP22gm1610008, to T.F.); JST PRESTO (JPMJPR22R1, to J.I.); JST CREST (JPMJCR20H4, to K.S.; JPMJCR20H8, to T.H.); JSPS KAKENHI Grant-in-Aid for Scientific Research C (22K07103, to T.I.); JSPS KAKENHI Grant-in-Aid for Scientific Research B (21H02736, to T.F.); JSPS KAKENHI Grant-in-Aid for Early-Career Scientists (22K16375, to H.N.; 20K15767 and 23K14526, J.I.); JSPS KAKENHI grant 20H05773 (to T.H.); JSPS Core-to-Core Program (A. Advanced Research Networks) (JPJSCCA20190008, to K.S.); JSPS Research Fellow DC2 (22J11578, to K.U.); JSPS Leading Initiative for Excellent Young Researchers (LEADER) (to T.I.); World-leading Innovative and Smart Education (WISE) Program 1801 from the Ministry of Education, Culture, Sports, Science and Technology (MEXT) (to N.N.); Research Support Project for Life Science and Drug Discovery [Basis for Supporting Innovative Drug Discovery and Life Science Research (BINDS)] from AMED under the Grant JP22ama121001 (to T.H.); The Cooperative Research Program (Joint Usage/Research Center program) of Institute for Life and Medical Sciences, Kyoto University (to K.S.); The Tokyo Biochemical Research Foundation (to K.S.); Takeda Science Foundation (to T.I.); Mochida Memorial Foundation for Medical and Pharmaceutical Research (to T.I.); The Naito Foundation (to T.I.); Shin-Nihon Foundation of Advanced Medical Research (to T.I.); Waksman Foundation of Japan (to T.I.); an intramural grant from Kumamoto University COVID-19 Research Projects (AMABIE) (to T.I.); the Ito Foundation Research Grant (to A.S.); International Joint Research Project of the Institute of Medical Science, the University of Tokyo (to T.I., T.F., and A.S.); the UK's Medical Research Council (to S.L.); and the project of National Institute of Virology and Bacteriology, Programme EXCELES, funded by the European Union, Next Generation EU (LX22NPO5103, to J.Z.).

## Author contributions

J.I. and S.L. performed phylogenetic analyses. J.I. performed statistical, modeling, and bioinformatics analyses. K.U., S.D., H.N., M.S., M.S.T.M.B., S.F., A.S., K.T., and T.I. performed the cell culture experiments. R.S., Y.I., T.T., I.K., K.Y., H.I., N.N., K.M., and T.F. performed animal experiments. L.W., M.T., Y.O., and S.T. performed histopathological analysis. J.Z. and G.S. performed the yeast surface display assay. S.D. and K.T. prepared AO-ALI and airway-on-a-chip systems. Y.Y. and T.N. performed the generation and provision of human iPSC-derived airway and alveolar epithelial cells. J.S., K.S.-T., and T.H. prepared BQ.1.1 S RBD and human ACE2. K.T.K., T.S., and T.H. determined the structure of the BQ.1.1 RBD and human ACE2 complex. H.A., M.N., K.S., and K.Y. performed viral genome sequencing analysis. J.K. contributed to clinical sample collection. J.I., A.S., K.M., K.T., S.T., T.F., T.I., and K.S. designed the experiments and interpreted the results. J.I. and K.S. wrote the original manuscript. All authors reviewed and proofread the manuscript. The Genotype to Phenotype Japan (G2P-Japan) Consortium contributed to the project administration.

## Competing interests

Y.Y. and T.N. are founders and shareholders of HiLung, Inc. Y.Y. is a coinventor of patents (PCT/JP2016/057254; "Method for inducing differentiation of alveolar epithelial cells", PCT/JP2016/059786, "Method of producing airway epithelial cells"). The other authors declare that no competing interests exist.

## Additional information

[1]Division of Systems Virology, Department of Microbiology and Immunology, The Institute of Medical Science, The University of Tokyo, Tokyo, Japan. [2]Department of Microbiology and Immunology, Faculty of Medicine, Hokkaido University, Sapporo, Japan. [3]Graduate School of Medicine, The University of Tokyo, Tokyo, Japan. [4]Division of Molecular Pathobiology, International Institute for Zoonosis Control, Hokkaido University, Sapporo, Japan. [5]Department of Biomolecular Sciences, Weizmann Institute of Science, Rehovot, Israel. [6]First Medical Faculty at Biocev, Charles University, Vestec-Prague, Czechia. [7]Laboratory of Medical Virology, Institute for Life and Medical Sciences, Kyoto University, Kyoto, Japan. [8]Center for iPS Cell Research and Application (CiRA), Kyoto University, Kyoto, Japan. [9]Department of Cancer Pathology, Faculty of Medicine, Hokkaido University, Sapporo, Japan. [10]Institute for Chemical Reaction Design and Discovery (WPI-ICReDD), Hokkaido University, Sapporo, Japan. [11]Medical Research Council-University of Glasgow Centre for Virus Research, Glasgow, UK. [12]Division of Risk Analysis and Management, International Institute for Zoonosis Control, Hokkaido University, Sapporo, Japan. [13]Division of Molecular Virology and Genetics, Joint Research Center for Human Retrovirus infection, Kumamoto University, Kumamoto, Japan. [14]Department of Clinical Pathology, Faculty of Medicine, Suez Canal University, Ismailia, Egypt. [15]Department of Veterinary Science, Faculty of Agriculture, University of Miyazaki, Miyazaki, Japan. [16]Graduate School of Medicine and Veterinary Medicine, University of Miyazaki, Miyazaki, Japan. [17]Department of Medicinal Sciences,

Graduate School of Pharmaceutical Sciences, Kyushu University, Fukuoka, Japan. [18]Institute for Genetic Medicine, Hokkaido University, Sapporo, Japan. [19]Division of International Research Promotion, International Institute for Zoonosis Control, Hokkaido University, Sapporo, Japan. [20]Tokyo Metropolitan Institute of Public Health, Tokyo, Japan. [21]HiLung, Inc, Kyoto, Japan. [22]Interpark Kuramochi Clinic, Utsunomiya, Japan. [23]Department of Global Health Promotion, Tokyo Medical and Dental University, Tokyo, Japan. [24]Center for Animal Disease Control, University of Miyazaki, Miyazaki, Japan. [25]One Health Research Center, Hokkaido University, Sapporo, Japan. [26]Institute for Vaccine Research and Development: HU-IVReD, Hokkaido University, Sapporo, Japan. [27]International Collaboration Unit, International Institute for Zoonosis Control, Hokkaido University, Sapporo, Japan. [28]AMED-CREST, Japan Agency for Medical Research and Development (AMED), Tokyo, Japan. [29]Laboratory of Virus Control, Research Institute for Microbial Diseases, Osaka University, Suita, Japan. [30]International Research Center for Infectious Diseases, The Institute of Medical Science, The University of Tokyo, Tokyo, Japan. [31]International Vaccine Design Center, The Institute of Medical Science, The University of Tokyo, Tokyo, Japan. [32]Graduate School of Frontier Sciences, The University of Tokyo, Kashiwa, Japan. [33]Collaboration Unit for Infection, Joint Research Center for Human Retrovirus infection, Kumamoto University, Kumamoto, Japan. [34]CREST, Japan Science and Technology Agency, Kawaguchi, Japan. [40]These authors contributed equally: Jumpei Ito, Rigel Suzuki, Keiya Uriu, Yukari Itakura, Jiri Zahradnik, Kanako Terakado Kimura, Sayaka Deguchi, Lei Wang, Spyros Lytras. ✉e-mail: hashiguchi.takao.1a@kyoto-u.ac.jp; tanaka@med.hokudai.ac.jp; fukut@pop.med.hokudai.ac.jp; ikedat@kumamoto-u.ac.jp; KeiSato@g.ecc.u-tokyo.ac.jp

## The Genotype to Phenotype Japan (G2P-Japan) Consortium

Saori Suzuki[2], Marie Kato[9], Zannatul Ferdous[9], Hiromi Mouri[9], Kenji Shishido[9], Naoko Misawa[1], Izumi Kimura[1], Yusuke Kosugi[1], Pan Lin[1], Mai Suganami[1], Mika Chiba[1], Ryo Yoshimura[1], Kyoko Yasuda[1], Keiko Iida[1], Naomi Ohsumi[1], Adam P. Strange[1], Daniel Sauter[1,35], So Nakagawa[36], Jiaqi Wu[36], Yukio Watanabe[8], Ayaka Sakamoto[8], Naoko Yasuhara[8], Yukari Nakajima[37], Hisano Yajima[37], Kotaro Shirakawa[37], Akifumi Takaori-Kondo[37], Kayoko Nagata[37], Yasuhiro Kazuma[37], Ryosuke Nomura[37], Yoshihito Horisawa[37], Yusuke Tashiro[37], Yugo Kawa[37], Takashi Irie[38], Ryoko Kawabata[38], Ryo Shimizu[13], Otowa Takahashi[13], Kimiko Ichihara[13], Chihiro Motozono[39], Mako Toyoda[39], Takamasa Ueno[39], Yuki Shibatani[15] & Tomoko Nishiuchi[15]

[35]University Hospital Tübingen, Tübingen, Germany. [36]Tokai University School of Medicine, Isehara, Japan. [37]Kyoto University, Kyoto, Japan. [38]Hiroshima University, Hiroshima, Japan. [39]Kumamoto University, Kumamoto, Japan.

