## [Peer Review File · Nature Communications]

Reviewers' Comments:

Reviewer #1:

Remarks to the Author:

Review: Convergent evolution of the SARS-CoV-2 Omicron subvariants leading to the emergence of BQ.1.1 variant

Summary

In this work, the authors combine the analysis of SARS-CoV-2 genomic data to in vitro and in vivo experimentations to provide a comprehensive description of the mutations emerged through convergent evolution within Omicron BQ1.1 lineage R346T, K444T and N460K. They start their analysis by describing convergent evolution within the Omicron family overall and use Bayesian hierarchical modeling on a set of 375,121 sequences from the UK to assess how substitutions at sites subject to convergent evolution in the S protein explain changes in virus fitness expressed as a relative effective reproduction number R_e , estimated from the growth rate of virus S haplotypes. They find that substitutions at five convergent sites are associated with a positive effect on R_e and estimate that throughout the evolutionary history of BA.5, multiple sublineages exhibited substitutions likely to give them a high fitness and in particular BQ1.1, for which fitness likely increased gradually with the sequential acquisition of mutations R346T, N460K, and K444T. The authors then characterize the effect of these three mutations in vitro and in vivo. They show that while the BQ1.1 S protein has a greater ability to escape from neutralization with BA.2 and BA.5 infection sera than BA.5 S, this property is not reproduced by any of the three convergent mutations alone. They find that BQ1.1 S exhibits increased ACE2 binding affinity, pseudovirus infectivity and fusogenicity, and that these properties can likely be attributed to the R346T and N460K substitutions. Finally, the authors evaluate the pathogenicity of BQ1.1 in hamsters and find that it is comparable to (if not lower than) that of BA.5 with that of BA.2 and BA.5.

General Comment

This work provides an in-depth analysis of the potential impact of three convergent mutations on the phenotype of SARS-CoV-2 at the population level (impact on virus fitness/ R_e) and at the individual (pathogenicity), cellular (infectivity) and molecular levels (binding affinity and fusogenicity). A great amount of work has been provided to ensure the comprehensive description of the impact of these mutations but I have several concerns that need to be addressed before the manuscript can be considered for publication.

Major Comments

Limitations on "virus fitness" estimates

In the second results section, the authors analyze the effect of substitutions at convergent sites on virus fitness based on the relative effective reproduction number estimated from a set of virus genomes from the UK. More specifically, the authors base their estimates on S haplotypes dynamics in a specific virus population. The observations made on "virus fitness" are thus specific to the study and may not be generalisable to other environments with different profiles of immunity in the human population or different social interaction levels/dynamics (containment measures, mask wearing). I recommend the authors insist on this limitation as this aspect of their modeling approach is quite different from that of Obermeyer et al. Also, in that section, they mention doing "ancestral fitness reconstruction" which I find confusing as this is rather an estimation of the fitness of ancestral sequences using their Bayesian hierarchical model. I would recommend finding another formulation.

Description of neutralization experiments

Throughout the result section describing the neutralization experiments (Immune resistance of BQ.1.1) the authors refer to the neutralizing effect of the sera as an "antiviral effect" and the escape of the S protein from neutralization as resistance. I would recommend using the term "neutralizing" instead of "antiviral" and "escape" instead of "resistance" to avoid any confusion for

the readers. Antiviral and resistance are more generally used to describe experiments involving antiviral drugs rather than neutralization experiments.

Theories explaining the discrepancy between viral fusogenicity and intrinsic pathogenicity in the Discussion section are too vague

Overall, both theories described at the end of the discussion to explain the discrepancy between viral fusogenicity and intrinsic pathogenicity are described in a very vague manner. Please clarify (see examples below).

-“acquired mutation(s) in the non-S region of viral genome that can attenuate viral pathogenicity and cancel the pathogenicity elevated by the higher fusogenicity compared with the parental BA.5”
I recommend the authors be more explicit and describe the mechanisms underlying the potential role of non-S mutations in decreasing pathogenicity because for now this part is very vague.

-“increased fusogenicity of viral S protein can reduce viral fitness in the human population because greater fusogenicity can result in elevating pathogenicity”
Similarly, please be explicit regarding the exact mechanisms underlying your scenario, what would be the link between higher fusogenicity and pathogenicity and what would be the link between higher pathogenicity and reduced fitness in the human population? Also, the observations made in this study on virus “fitness” are specific to a given population (UK sequences), fitness may vary in a population with a different immunological background.

-“antigenic escape can augment viral pathogenicity” again, be more explicit so the reader can have the full mechanism in mind.

-“More importantly, this theory also predicts that there is a limitation to increase viral pathogenicity. Together with our observations, it might be possible to assume that the pathogenicity of Omicron lineage already reaches a plateau.”
Again here, please explain what would be the underlying mechanism.

Language

I am not a native speaker and I understand the challenge of writing in a different language but the language needs to be improved because it is sometimes confusing in the current version of the manuscript.

Minor comments

L124: “evolved to increase intrinsic pathogenicity”. Increased pathogenicity is not a goal for the virus. Evolution is the result of a random mutational process and some degree of selection (variable depending on the population and on the environment in which the virus replicates and spreads). As it is not obvious that increased intrinsic pathogenicity would necessarily be favored by selection, I recommend that the authors be cautious regarding the formulations they use here and more generally, regarding the way they describe evolutionary processes throughout the manuscript.

L168-178 “Consistent with our previous study, the L452 residue in BA.2 showed the highest substitution frequency in that lineage (Fig. 1c–e)...9.4-times higher than those in BA.2, respectively.”

These are a quantitative statements, so please provide numbers to support them

L212 “to show higher R e values”

Here please provide numbers to support your statement.

L264-266: “suggesting that multiple substitutions cooperatively contribute to the immune resistance of BQ.1.1 to breakthrough BA.2 infection sera.”

These results are interesting, have you tested combinations of mutations ?

L290-293: "As shown in Fig. 2d, the cross-reactivity of each Omicron subvariant was well correlated to their phylogenetic relationship (Fig. 1a)"

If you used a correlation test please describe it and provide the results alternatively, if you are simply mentioning a correspondence between phylogenetic relationships and antigenic cartography rephrase.

L418 please define "intrinsic pathogenicity" clearly, as this term is often used in your study.

L442-445 "Moreover, the reconstruction of ancestral viral fitness suggests that the ancestral lineage of BQ.1.1 has increased viral fitness by acquiring substitutions at the convergent sites in a stepwise manner (Fig. 1i)."

I would rephrase here: ancestral sequence reconstruction suggests that mutations increasing "virus fitness" emerged within the BQ1.1 sublineage in a stepwise manner. If I understood correctly, you do not directly reconstruct fitness but reconstruct sequences and then estimate their fitness based on your model.

L528-534 "The emergence of SARS-CoV-2 variants with increased intrinsic pathogenicity may not be so critical for the immunized segment of the population. However, the variants bearing greater intrinsic pathogenicity can be a meaningful risk for people who do not have anti-SARS-CoV-2 immunity, most conspicuously the unvaccinated population, including children. Therefore, continued in-depth viral genomic surveillance and real-time evaluation of the risk of newly emerging SARS-CoV-2 variants should be crucial."

This last paragraph contradicts what has been mentioned just above, if immune escape is responsible for higher pathogenicity then, it is a concern for the vaccinated part of the population. Also, SARS-CoV-2 is notoriously more severe in older individuals rather than children so I feel the second to last statement should be rephrased.

L949-950 "Internal nodes with substitution probabilities above or equal to 0.5"

Have you tested how sensitive your results were to this threshold (e.g. increasing it to 0.7) ?

Reviewer #2:

Remarks to the Author:

This is quite an impressive analysis, with lots of experimental evidence supporting the authors specific claims, and to some extent, the general descriptions/explanations of the evolution of Omicron lineages observed over the past year. I do recommend publication of this paper in Nature Communications. I have a few comments below on interpretation and presentation, and some of these do need to be incorporated before the final ms is accepted.

1. One very important point is that our ability to infer R_e from sequence data alone is not quite as good as we think. The real quantity we are inferring is $w =$ "sampled at a higher proportion in later generations than in earlier generations" and this w does indeed get us a good approximation of relative fitness, that is, the relative fitness of the later genotype compared with the earlier genotype it replaced. It is hard to translate this into a fitness measure comparable across all genotypes because all pairwise replacements are not observed. For example, g_2 might replace g_1 at a certain rate, and this gives us a relative fitness w_{21} of g_2 over g_1 . Then, a lot of minor variants might pass through our surveillance system, until g_3 begins to dominate, and then g_4 replaces g_3 . This gives us a replacement rate of g_4 over g_3 , which gives us w_{43} , the relative fitness of g_4 over g_3 . But, with this data set, we don't get a clean replacement picture of g_3 over g_2 , and the relative fitness w_{32} is hard to estimate. The estimation procedure does estimate this quantity, probably with a lot of uncertainty. But the resulting pattern of fitness evolution in Figure 1h is unlikely to be this clean and simple.

A second reason the pattern of fitness evolution may not be so clean and simple is that the environment is changing the entire time these samples are being collected. The assays in Figures 2 and 3 are done in identical environments and allow us to compute relative fitnesses. But, the

surveillance sampling done for Figure 1 (the Obermeyer approach) doesn't measure any phenotype, it just counts genotypes, and it assumes that a change in relative genotype proportions must have occurred in a constant environment, therefore it must be appropriate to use this frequency change to measure fitness. The Obermeyer authors (I am not one of them) acknowledge this limitation. Please read pages 6-7 of their supplement carefully. They mention all the dynamics that are not accounted for in the inference: dynamics due to (a) biased sampling in surveillance, (b) changes in population density as the virus moves from place to place, (c) changes in human contact patterns as the epidemic proceeds, (d) difference in background immunity when comparing one genotype replacement with another genotype replacement, and possibly more.

Finally, for this R_e measurement: you cannot disentangle the effects of increased transmissibility from increased immune escape, because all mutations are in the spike protein. If g2 replaced g1, the Obermeyer R_e estimate cannot tell you if this replacement occurred because of (1) immune escape, (2) different transmissibility, or (3) a combination of the two.

2. The authors present a story in the introduction and discussion of why the convergent evolution proceeded the way that it did. The reader needs a summary of how strong the evidence base behind this story is. Something like:

Strongest evidence: presence of homoplasies, immune escape assays, cell-culture growth kinetics

Strong evidence: R_e estimates

Weaker evidence: hamster phenotypes (bc these were not done for individual mutations)

The authors may disagree with my assessment above. This is fine. But, as you're telling the story of these convergent mutations, and as you observe changes in ACE2 binding, in vitro growth, pathogenicity in hamsters, and immune escape: tell the reader which phenotypic changes are supported by the strongest evidence. It seems that the simple presence of all these homoplasies is very good evidence for convergent evolution. The immune assays and in vitro assays on specific genotypes certainly give good phenotypic descriptions of all the genotypes. And (it seems to me) the R_e estimates and the virulence phenotypes show weaker evidence of phenotypic change.

3. Related to point 2 above: I don't think the relationships among these phenotypes are very strong. Despite the literature cited in para 3 of the introduction, I don't think we have good evidence that ACE2 binding evolves and that this always results in increased/decreased pathogenicity. I don't think we have good evidence that immune escape occurs and that this always results in increased/decreased pathogenicity. Your picture of phenotypic evolution is quite solid. But, I don't think it is broadly accepted that the associations among these phenotypes always fall in one direction or another.

4. Please use the term "immune escape" not "immune resistance"

5. Please list the five mutations in the abstract (word count permitting), and tell the reader whether K444T follows the same pattern as the other four. It seemed that this mutation had a different ACE2 phenotype from the others and that the reason for its evolutionary path may have been different. But, this part of the paper was sufficiently complex that I could be misreading something here.

6. The fourth paragraph of the introduction needs work. I think the authors are trying to say something like my re-write below:

Until the emergence of BA.5, newly emerging SARS-CoV-2 variants were outcompeting previously dominant variants on a several-month replacement cycle. However, as of November 2022, although a variety of Omicron subvariants (including BA.2.75) had emerged after BA.5, none of them had successfully outcompeted BA.5. Instead of the fixation of a single dominant SARS-CoV-2 variant, recently emerged Omicron subvariants have co-circulated due to a process of convergent evolution that may have established equivalent fitnesses among these lineages. Most of these co-circulating variants acquired substitutions at the same sites of S, namely, R346, K444, L452,

N460, and F486.

If this is correct, please feel free to directly use the text above. This is a referee editorial suggestion that you are free to use as is.

7. Lit review: a lot has been written on the observed virulence/transmission characteristics of H5N1 influenza viruses, and the potential virulence/transmission characteristics of SARS-CoV-2. It would be worth citing several of the key papers here to give appropriate background. The Sasaki-Lion-Boots paper is not the last word on this, and there were many papers before that have addressed this foundational question in pathogen evolution. Indeed, is immune-escape a necessary mediator of the virulence/transmission relationship? If this result is so generalizable why have H3N2 influenza viruses not become more transmissible and more virulent over the past 55 years? Why have noroviruses not become more virulent?

8. Verb tenses in the discussion need attention

9. Last paragraph in the discussion is not very good. The weakest evidence you have is for evolution of pathogenicity. Strongest evidence is for convergent evolution itself.

Reviewer #3:

Remarks to the Author:

Ito and colleagues have provided a detailed investigation of convergent evolution in the spike protein of SARS-CoV-2 Omicron subvariants and their functional impacts on viral behaviors and pathogenesis. This is an intriguing study given the integration of modelling approaches to estimating viral fitness impacts for spike protein mutations. Importantly, this was integrated with functional investigations using both pseudotyped viruses and wild-type viruses. This is an impactful study for our understanding of viral evolution.

However, there are some key points of consideration for the authors that should be addressed.

These are detailed below:

-Some of the grammar is confusing in sections, such as lines 260-262 with "either of these three substitutions". This becomes clear in the following sentences (cooperative contributions vs individual activities); however, this could be more precisely worded up front for reader clarity.

-in lines 269-270: A question here is how the sera from different breakthrough infection cases is being normalized or standardized. For example, in an instance where an individual has a below-average response to vaccination in regard to antibody generation, how would this sera compare to that from a strong or average responder?

-in lines 287-288: How well can hamster sera be used for validation here given that the neutralization activities of BA.2, BA.5 and BA.2.75 were largely completely negated for other subvariants. How is this reflective of human sera and does this have implications for the robustness of these types of data?

-in the ACE2 Binding Affinity section, the following consideration should be noted. The definition of infection relies on both entry and multiplication of a pathogen. Are these replication-competent pseudotypes? If not, this is an assessment of binding and entry, which may correlate with infectivity, but that needs to be addressed with replication kinetics.

-lines 356-361: It would be prudent for the authors to provide some comment on the differences in replication kinetic dynamics between the iPSC-derived airway ECs and the ALI NHBEs.

-lines 363-371: The discussion on barrier disruption should be verified by a secondary or alternative method. Viruses can move from epithelial to endothelial barriers without appreciable differences in cell-cell contacts as demonstrated by TEER or ECIS experiments. Disruption could be addressed through these methods, fluorescein-based experiments, or microscopy

-lines 374-375: There is no description provided here of what was inoculated, how, route, etc. There should be some description provided here for species and route.

-in line 398, the authors mention "spreading efficacy". What is spreading efficacy and how is this assessed? Do the authors mean viral dissemination?

Reviewer #4:

Remarks to the Author:

In this study, Ito et al investigate BQ.1.1, a recently emerged variant of SARS-CoV-2 which has expanded to become the currently predominating strain globally (with BQ.1). In particular, this variant is notable for descending from, and displacing, BA.5, which was the previously globally dominant variant. Given this dominance, studies of this variant are warranted, in order to understand its virological characteristics as well as its evolutionary basis. While there have been a number of other studies that have been published on the former, the focus on evolution, which is the main topic of this manuscript, has not been reported to this reviewer's knowledge, and this study is therefore an important contribution to the field.

The authors identify that many BA.5 descendants harbor mutations within the same residues in the spike protein, including BQ.1.1, and they therefore examine these residues using a phylogenetic approach, finding that several sites confer a fitness advantage for the virus. Notably, these analyses suggest that the mutations that led to BQ.1.1 are K444T, N460K, and R346T in the spike, in this order. The authors then test the virological characteristics of BQ.1.1. They first examine the serum neutralization of BQ.1.1 and the aforementioned three point mutations in the background of BA.5, finding that BQ.1.1 is the most resistant and that its serum resistance is not attributable to a particular single mutation. They next examine the ACE2 affinity, finding that the heightened affinity of BQ.1.1 is attributable to N460K, and that this correlated with its increased infectivity and fusogenicity. BQ.1.1 displayed greater growth kinetics in some models, whereas in others it displayed no difference or had decreased growth. Finally, in an in vivo hamster model, BQ.1.1 was comparable to BA.5 in terms of pathogenicity and spreading efficiency.

This study is of high quality and well-written, and the data are presented in a clear manner. The Sato Lab and their consortia members have made many contributions to the SARS-CoV-2 field and this study continues their outstanding work. The manuscript is appropriate for this journal and should be accepted, but this reviewer has some suggestions that may improve the manuscript, outlined below.

Major comments:

- While the focus of this study is understandably the spike protein, BQ.1.1 carries several additional mutations in other genes (as noted by the authors in Fig 1b). A thorough virological analysis of these mutations is certainly outside the scope of this work, but where do these mutations rank within the phylogenetic analyses that were conducted? That is, are these mutations partially involved in the evolutionary trajectory of and rise of BQ.1.1, or is it only the spike mutations that are important?
- An important and unique takeaway of this study is that BQ.1.1 arose through K444T, N460K, and R346T in order, according to the phylogenetic analyses presented in Fig 1. This is of course of importance as it furthers our understanding of how SARS-CoV-2 is evolving in the population, and why it may be doing so in this particular manner. However, the subsequent experiments and data seem to shy away from this evolutionary perspective. This reviewer suggests that some experimental evidence for this proposed evolutionary route would greatly strengthen this manuscript and differentiate it from the others that have been published, as similar data to what are currently presented in Figs. 2-3 have been described by other groups. One suggestion would be to repeat some of the experiments in Fig 2 and in Fig 3 with the inclusion of the combinatorial mutants (e.g., K444T + N460K, K444T + R346T, N460K + R346T). This should not be too much additional work, but would serve to validate the phylogenetic analysis. It may also shed light on the role of K444T, which, in the currently presented dataset, does not seem to have a specific functional role, yet is suggested to have arose first in the rise of BQ.1.1.
- In a similar vein, are there any thoughts on how the emergence of BQ.1.1 mirrors what occurred previously with other SARS-CoV-2 variants, and whether there seems to be a particular type of route that the virus favors? That is, do certain mutations that confer certain phenotypes occur at a certain point in the virus' evolution? For example, the R346T mutation is reminiscent of the R346K mutation that arose in BA.1 to form BA.1.1, but is this coincidence or are we seeing a particular

pattern?

Minor comments:

- Line 145: "principals" should be "principles"
- Line 193: Why were Omicron sequences only used from the UK and not globally?
- Line 371: "fusogenic" should be "fusogenicity"
- Line 1498: There seems to be an error with this reference
- Fig 1i: The x-axis has the label "2023/9" on the right panel; should this not be "2022"?

Reviewer #1 (Remarks to the Author):

Review: Convergent evolution of the SARS-CoV-2 Omicron subvariants leading to the emergence of BQ.1.1 variant

Summary

In this work, the authors combine the analysis of SARS-CoV-2 genomic data to in vitro and in vivo experimentations to provide a comprehensive description of the mutations emerged through convergent evolution within Omicron BQ1.1 lineage R346T, K444T and N460K. They start their analysis by describing convergent evolution within the Omicron family overall and use Bayesian hierarchical modeling on a set of 375,121 sequences from the UK to assess how substitutions at sites subject to convergent evolution in the S protein explain changes in virus fitness expressed as a relative effective reproduction number R_e , estimated from the growth rate of virus S haplotypes. They find that substitutions at five convergent sites are associated with a positive effect on R_e and estimate that throughout the evolutionary history of BA.5, multiple sublineages exhibited substitutions likely to give them a high fitness and in particular BQ1.1, for which fitness likely increased gradually with the sequential acquisition of mutations R346T, N460K, and K444T. The authors then characterize the effect of these three mutations in vitro and in vivo. They show that while the BQ1.1 S protein has a greater ability to escape from neutralization with BA.2 and BA.5 infection sera than BA.5 S, this property is not reproduced by any of the three convergent mutations alone. They find that BQ1.1 S exhibits increased ACE2 binding affinity, pseudovirus infectivity and fusogenicity, and that these properties can likely be attributed to the R346T and N460K substitutions. Finally, the authors evaluate the pathogenicity of BQ1.1 in hamsters and find that it is comparable to (if not lower than) that of BA.5 with that of BA.2 and BA.5.

General Comment

This work provides an in-depth analysis of the potential impact of three convergent mutations on the phenotype of SARS-CoV-2 at the population level (impact on virus fitness/ R_e) and at the individual (pathogenicity), cellular (infectivity) and molecular levels (binding affinity and fusogenicity). A great amount of work has been provided to ensure the comprehensive description of the impact of these mutations but I have several concerns that need to be addressed before the manuscript can be considered for publication.

Our reply:

We would like to thank the reviewer for kindly and deeply reviewing our work. According to the reviewer's suggestions, we have modified our study to the best of our ability.

Major Comments

Limitations on “virus fitness” estimates

In the second results section, the authors analyze the effect of substitutions at convergent sites on virus fitness based on the relative effective reproduction number estimated from a set of virus genomes from the UK. More specifically, the authors base their estimates on S haplotypes dynamics in a specific virus population. The observations made on “virus fitness” are thus specific to the study and may not be generalisable to other environments with different profiles of immunity in the human population or different social interaction levels/dynamics (containment measures, mask wearing). I recommend the authors insist on this limitation as this aspect of their modeling approach is quite different from that of Obermeyer et al.

Our reply:

We agree that viral fitness would change according to host factors such as herd immunity in the human population or different social interaction levels, and therefore viral fitness estimated in the present study, using genome surveillance data from the UK for a specific period, does not necessarily generalize to other populations/countries. We have now added a limitations paragraph in the **Discussion** section where we extensively clarify this point. We also specified all the differences between our model and the model implemented by Obermeyer et al. in our text.

To evaluate how robust the estimation of viral fitness by our hierarchal model is to data from different populations (i.e., countries), we additionally analyzed the genome surveillance data from 19 more countries providing a higher number of sequences for our model (**Fig. 2d,e and Extended Data Fig. 2; Fig. R1**): First, we predicted the relative R_e of each S haplotype in each county using the hierarchal model trained by the genome surveillance data from the UK. Subsequently, we compared the predicted R_e of each S haplotype in each country with the R_e estimated by a simple multinomial logistic model based on the genome surveillance data from each country. We confirmed that the predicted R_e was highly concordant to the estimated R_e (adjusted $R^2 > 0.9$) in most countries investigated. Lower R^2 for some countries could be explained by the presence of a few outlier S haplotypes. Furthermore, the trained model successfully predicted the higher R_e of XBB (a recombinant lineage between two highly divergent BA.2 variants [doi: <https://doi.org/10.1101/2022.12.27.521986>]) and BA.2.3.20 (a highly diversified BA.2 sublineage harboring 10 substitutions compared to BA.2 in S1 subunit including L452M, K444R, and N460K [<https://github.com/cov-lineages/pango-designation/issues/1013>]) even though these variants were not included in the training data (**Fig. 2d**). Together, these results suggest that our model captured the key information of the fitness landscape underlying the convergent evolution of Omicron lineages observed worldwide in late 2022.

Fig. R1. Prediction of the relative R_e of S haplotypes in each country using the model trained on the UK's data.

a, The predicted R_e of S haplotypes and R_e estimated by a multiple logistic model based on each country's data were compared. Dot size indicates the number of sequences of each haplotype. The dotted line denotes a line with slope of 1 and intercept of 0. The 20 countries with the highest number of sequences were analyzed.

b, Adjusted R^2 value of the prediction in each country. Bar color indicates total number of sequences in each country.

Also, in that section, they mention doing “ancestral fitness reconstruction” which I find confusing as this is rather an estimation of the fitness of ancestral sequences using their Bayesian hierarchical model. I would recommend finding another formulation.

Our reply:

We apologize for the misleading expression. We modified this formulation through the revised manuscript according to the reviewer's suggestion (lines 252–255, page 9; lines 555–559, page 17).

Description of neutralization experiments

Throughout the result section describing the neutralization experiments (Immune resistance of BQ.1.1) the authors refer to the neutralizing effect of the sera as an “antiviral effect” and the escape of the S protein from neutralization as resistance. I would recommend using the term “neutralizing” instead of “antiviral” and “escape” instead of “resistance” to avoid any confusion for the readers. Antiviral and resistance are more generally used to describe experiments involving antiviral drugs rather than neutralization experiments.

Our reply:

We modified the revised manuscript according to the reviewer's suggestion.

Theories explaining the discrepancy between viral fusogenicity and intrinsic pathogenicity in the Discussion section are too vague

Overall, both theories described at the end of the discussion to explain the discrepancy between viral fusogenicity and intrinsic pathogenicity are described in a very vague manner. Please clarify (see examples below).

-“acquired mutation(s) in the non-S region of viral genome that can attenuate viral pathogenicity and cancel the pathogenicity elevated by the higher fusogenicity compared with the parental BA.5”

I recommend the authors be more explicit and describe the mechanisms underlying the potential role of non-S mutations in decreasing pathogenicity because for now this part is very vague.

-“increased fusogenicity of viral S protein can reduce viral fitness in the human population because greater fusogenicity can result in elevating pathogenicity”

Similarly, please be explicit regarding the exact mechanisms underlying your scenario, what would be the link between higher fusogenicity and pathogenicity and what would be the link between higher pathogenicity and reduced fitness in the human population? Also, the observations made in this study on virus “fitness” are specific to a given population (UK sequences), fitness may vary in a population with a different immunological background.

-“antigenic escape can augment viral pathogenicity” again, be more explicit so the reader can have the full mechanism in mind.

-“More importantly, this theory also predicts that there is a limitation to increase viral pathogenicity. Together with our observations, it might be possible to assume that the pathogenicity of Omicron lineage already reaches a plateau.”
Again here, please explain what would be the underlying mechanism.

Our reply:

We thank the reviewer for pointing this out and agree that our discussion on the discrepancy between viral fusogenicity and intrinsic pathogenicity was rather speculative. We have now removed this specific paragraph from the **Discussion** section and only include improved and better substantiated discussion points in the revised manuscript.

Language

I am not a native speaker and I understand the challenge of writing in a different language but the language needs to be improved because it is sometimes confusing in the current version of the manuscript.

Our reply:

We apologize for our insufficient English ability causing confusion. The revised manuscript has been proofread by multiple native English speakers. We hope the revised manuscript has improved clarity.

Minor comments

L124: “evolved to increase intrinsic pathogenicity”. Increased pathogenicity is not a goal for the virus. Evolution is the result of a random mutational process and some degree of selection (variable depending on the population and on the environment in which the virus replicates and spreads). As it is not obvious that increased intrinsic pathogenicity would necessarily be favored by selection, I recommend that the authors be cautious regarding the formulations they use here and more generally, regarding the way they describe evolutionary processes throughout the manuscript.

Our reply:

We agree with this comment and have modified expressions throughout the revised manuscript.

L168-178 “Consistent with our previous study, the L452 residue in BA.2 showed the highest substitution frequency in that lineage (Fig. 1c–e)...9.4-times higher than those in BA.2, respectively.”

These are a quantitative statements, so please provide numbers to support them

Our reply:

We have added the quantitative information requested (line 171–181, page 7).

L212 “to show higher R_e values”

Here please provide numbers to support your statement.

Our reply:

We have added the relative R_e value for BQ.1.1 in the revised manuscript (line 222, page 8).

L264-266: “suggesting that multiple substitutions cooperatively contribute to the immune resistance of BQ.1.1 to breakthrough BA.2 infection sera.”

These results are interesting, have you tested combinations of mutations ?

Our reply:

We have not tested combinations of mutations. Combinations would be interesting, but there are 8 patterns of combinations in this case, which would be technically hard and time-consuming to perform.

L290-293: “As shown in Fig. 2d, the cross-reactivity of each Omicron subvariant was well correlated to their phylogenetic relationship (Fig. 1a)”

If you used a correlation test please describe it and provide the results alternatively, if you are simply mentioning a correspondence between phylogenetic relationships and antigenic cartography rephrase.

Our reply:

We have rephrased this sentence in accordance with your comment (line 319, page 11).

L418 please define “intrinsic pathogenicity” clearly, as this term is often used in your study.

Our reply:

In the revised manuscript, intrinsic pathogenicity has been defined in the Introduction section (line 145, page 5).

L442-445 “Moreover, the reconstruction of ancestral viral fitness suggests that the ancestral lineage of BQ.1.1 has increased viral fitness by acquiring substitutions at the convergent sites in a stepwise manner (Fig. 1i).”

I **would** rephrase here: ancestral sequence reconstruction suggests that mutations increasing “virus fitness” emerged within the BQ1.1 sublineage in a stepwise manner. If I understood correctly, you do not directly reconstruct fitness but reconstruct sequences and then estimate their fitness based on your model.

Our reply:

We agree with this comment and have rephased sentences related to the point above throughout the manuscript (lines 252–255, page 9; lines 555–559, page 17).

L528-534 “The emergence of SARS-CoV-2 variants with increased intrinsic pathogenicity may not be so critical for the immunized segment of the population. However, the variants bearing greater intrinsic pathogenicity can be a meaningful risk for people who do not have anti-SARS-CoV-2 immunity, most conspicuously the unvaccinated population, including children. Therefore, continued in-depth viral genomic surveillance and real-time evaluation of the risk of newly emerging SARS-CoV-2 variants should be crucial.”

This last paragraph contradicts what has been mentioned just above, if immune escape is responsible for higher pathogenicity then, it is a concern for the vaccinated part of the population. Also, SARS-CoV-2 is notoriously more severe in older individuals rather than children so I feel the second to last statement should be rephrased.

Our reply:

According to this comment and the comment from **Reviewer #2**, we have modified the conclusion paragraph in the **Discussion** section: in the revised manuscript, we mainly focus on our findings regarding the importance of convergent evolution in the emergence of BQ.1.1 rather than those on its intrinsic pathogenicity.

L949-950 “Internal nodes with substitution probabilities above or equal to 0.5”
Have you tested how sensitive your results were to this threshold (e.g. increasing it to 0.7) ?

Our reply:

Regarding the threshold of minimum probability to define that a substitution exists at each internal node, we performed a sensitivity analysis (**Extended Data Fig. 1b** and **Fig. R2** below). We confirmed that the number of acquisition events of substitutions is robust to this threshold in the range of 0.4–0.9.

Fig. R2. A sensitivity analysis on the number of detected substitution events.

The minimum probability to define that a substitution exists at each internal node was changed from 0.4 to 0.9 in increments of 0.1, and the number of detected substitution events was counted.

Reviewer #2 (Remarks to the Author):

This is quite an impressive analysis, with lots of experimental evidence supporting the authors specific claims, and to some extent, the general descriptions/explanations of the evolution of Omicron lineages observed over the past year. I do recomended publication of this paper in Nature Communications. I have a few comments below on interpretation and presentation, and some of these do need to be incorporated before the final ms is accepted.

Our reply:

First of all, we are very happy to hear your positive feedback about our study. The revised manuscript has been modified according to reviewer suggestions.

1. One very important point is that our ability to infer R_e from sequence data alone is not quite as good as we think. The real quantity we are inferring is $w =$ "sampled at a higher proportion in later generations than in earlier generations" and this w does indeed get us a good approximation of relative fitness, that is, the relative fitness of the later genotype compared with the earlier genotype it replaced. It is hard to translate this into a fitness measure comparable across all genotypes because all pairwise replacements are not observed. For example, g_2 might replace g_1 at a certain rate, and this gives us a relative fitness w_{21} of g_2 over g_1 . Then, a lot of minor variants might pass through our surveillance system, until g_3 begins to dominate, and then g_4 replaces g_3 . This gives us a replacement rate of g_4 over g_3 , which gives us w_{43} , the relative fitness of g_4 over g_3 . But, with this data set, we don't get a clean replacement picture of g_3 over g_2 , and the relative fitness w_{32} is hard to estimate. The estimation procedure does estimate this quantity, probably with a lot of uncertainty. But the resulting pattern of fitness evolution in Figure 1h is unlikely to be this clean and simple.

Our reply:

We agree with this comment and now carefully discuss these limitations in an extensive paragraph included in **Discussion** (lines 642–665, page 19).

A second reason the pattern of fitness evolution may not be so clean and simple is that the environment is changing the entire time these samples are being collected. The assays in Figures 2 and 3 are done in identical environments and allow us to compute relative fitnesses. But, the surveillance sampling done for Figure 1 (the Obermeyer approach) doesn't measure any phenotype, it just counts genotypes, and it assumes that a change in relative genotype proportions must have occurred in a constant environment, therefore it must be appropriate to use this frequency change to measure fitness. The Obermeyer authors (I am not one of them) acknowledge this limitation. Please read pages 6-7 of their supplement carefully. They mention all the dynamics that are not accounted for in the inference: dynamics due to (a) biased sampling in surveillance, (b) changes

in population density as the virus moves from place to place, (c) changes in human contact patterns as the epidemic proceeds, (d) difference in background immunity when comparing one genotype replacement with another genotype replacement, and possibly more.

Our reply:

We agree that viral fitness depends on environmental factors and have clarified this point in the limitation paragraph in the **Discussion** section.

Fig. R1. Prediction of the relative R_e of S haplotypes in each country using the model trained on the UK's data.

a, The predicted R_e of S haplotypes and R_e estimated by a multiple logistic model based on each country's data were compared. Dot size indicates the number of sequences of each haplotype. The dotted line denotes a line with slope of 1 and intercept of 0. The 20 countries with the highest number of sequences were analyzed.

b, Adjusted R^2 value of the prediction in each country. Bar color indicates total number of sequences in each country.

To evaluate how robust the estimation of viral fitness by our hierarchal model is to the different environments (e.g., countries), we additionally analyzed the genome surveillance data from 19 more countries providing a higher number of sequences for our model (**Fig. 2d,e and Extended Data Fig. 2; Fig. R1** above): First, we predicted the relative R_e of each S haplotype in each country using the hierarchal model trained by the genome surveillance data from the UK. Subsequently, we compared the predicted R_e of each S haplotype in each country with the R_e estimated by a simple multinomial logistic model based on the genome surveillance data from each country. We confirmed that the predicted R_e was highly concordant to the estimated R_e (adjusted $R^2 > 0.9$) in most countries investigated. Lower R^2 for some countries could be explained by the presence of a few outlier S haplotypes. Furthermore, the trained model successfully predicted the higher R_e of XBB (a recombinant lineage between two highly divergent BA.2 variants [doi: <https://doi.org/10.1101/2022.12.27.521986>]) and BA.2.3.20 (a highly diversified BA.2 sublineage harboring 10 substitutions compared to BA.2 in S1 subunit including L452M, K444R, and N460K [<https://github.com/cov-lineages/pango-designation/issues/1013>]) even though these variants were not included in the training data (**Fig. 2d**). Together, these results suggest that our model captured the key information of the fitness landscape underlying the convergent evolution of Omicron lineages observed worldwide in late 2022.

Finally, for this R_e measurement: you cannot disentangle the effects of increased transmissibility from increased immune escape, because all mutations are in the spike protein. If g2 replaced g1, the Obermeyer R_e estimate cannot tell you if this replacement occurred because of (1) immune escape, (2) different transmissibility, or (3) a combination of the two.

Our reply:

We agree that R_e viral fitness can be attributed to both/either i) immune escape ability and ii) pure transmissibility. In the revised manuscript, we clarified this point in the **Results** section (where R_e is defined) and a paragraph in the **Discussion** section.

2. The authors present a story in the introduction and discussion of why the convergent evolution proceeded the way that it did. The reader needs a summary of how strong the evidence base behind this story is. Something like:

Strongest evidence: presence of homoplasies, immune escape assays, cell-culture growth kinetics

Strong evidence: R_e estimates

Weaker evidence: hamster phenotypes (bc these were not done for individual mutations)

The authors may disagree with my assessment above. This is fine. But, as you're telling the story of these convergent mutations, and as you observe changes in ACE2 binding, in vitro growth, pathogenicity in hamsters, and immune escape: tell the reader which phenotypic changes are supported by the strongest evidence. It seems that the simple presence of all these homoplasies is very good evidence for convergent evolution. The immune assays and in vitro assays on specific genotypes certainly give good phenotypic descriptions of all the genotypes. And (it seems to me) the R_e estimates and the virulence phenotypes show weaker evidence of phenotypic change.

Our reply:

This is certainly of much importance for revising our initial manuscript. We appreciate this important suggestion, and agree with you on the clear assessment of the evidence we present. Our revised **Discussion** section hopefully better reflects this assessment. Specifically, we i) removed sections where conflicting statements were made regarding BQ.1.1's pathogenicity; ii) removed emphasis from the hamster experiment results; iii) explicitly described the limitations of the R_e estimate analysis in an added paragraph, explaining how context-specific these results can be; and iv) emphasized the importance of the level of convergence observed in these five substitutions and the implications this process can have in understanding SARS-CoV-2's evolution.

3. Related to point 2 above: I don't think the relationships among these phenotypes are very strong. Despite the literature cited in para 3 of the introduction, I don't think we have good evidence that ACE2 binding evolves and that this always results in increased/decreased pathogenicity. I don't think we have good evidence that immune escape occurs and that this always results in increased/decreased pathogenicity. Your picture of phenotypic evolution is quite solid. But, I don't think it is broadly accepted that the associations among these phenotypes always fall in one direction or another.

Our reply:

We agree and have removed the paragraph discussing the hypothesis on the relationship of immune escaping, pathogenicity, and viral fitness in the **Discussion** from the revised manuscript. Also, we removed the claims on the association of the evolution of ACE2 binding ability and viral intrinsic pathogenicity.

4. Please use the term "immune escape" not "immune resistance"

Our reply:

Fixed.

5. Please list the five mutations in the abstract (word count permitting), and tell the reader whether K444T follows the same pattern as the other four. It seemed that this mutation had a different ACE2 phenotype from the others and that the reason for its evolutionary path may have been different. But, this part of the paper was sufficiently complex that I could be misreading something here.

Our reply:

According to this suggestion, we have modified the Abstract (lines 90-104, page 4).

6. The fourth paragraph of the introduction needs work. I think the authors are trying to say something like my re-write below:

Until the emergence of BA.5, newly emerging SARS-CoV-2 variants were outcompeting previously dominant variants on a several-month replacement cycle. However, as of November 2022, although a variety of Omicron subvariants (including BA.2.75) had emerged after BA.5, none of them had successfully outcompeted BA.5. Instead of the fixation of a single dominant SARS-CoV-2 variant, recently emerged Omicron subvariants have co-circulated due to a process of convergent evolution that may have established equivalent fitnesses among these lineages. Most of these co-circulating variants acquired substitutions at the same sites of S, namely, R346, K444, L452, N460, and F486.

If this is correct, please feel free to directly use the text above. This is a referee editorial suggestion that you are free to use as is.

Our reply:

Greatly appreciated! In the revised manuscript, we have directly used the text prepared by the reviewer (lines 125-133, page 5).

7. Lit review: a lot has been written on the observed virulence/transmission characteristics of H5N1 influenza viruses, and the potential virulence/transmission characteristics of SARS-CoV-2. It would be worth citing several of the key papers here to give appropriate background. The Sasaki-Lion-Boots paper is not the last word on this, and there were many papers before that have addressed this foundational question in pathogen evolution. Indeed, is immune-escape a necessary mediator of the virulence/transmission relationship? If this result is so generalizable why have H3N2 influenza viruses not become more transmissible and more virulent over the past 55 years? Why have noroviruses not become more virulent?

Our reply:

We agree that relation between viruses' virulence and transmissibility is much more nuanced than depicted in our original submission. We have decided to remove the paragraph discussing the hypothesis on the relationship of immune escaping, pathogenicity, and viral fitness in the **Discussion** in the revised manuscript and focus on topics better elucidated by the presented results, such as the importance of convergence in SARS-CoV-2's evolution.

8. Verb tenses in the discussion need attention

Our reply:

We have modified them to the best of our ability.

9. Last paragraph in the discussion is not very good. The weakest evidence you have is for evolution of pathogenicity. Strongest evidence is for convergent evolution itself.

Our reply:

In accordance with this comment, we have modified the conclusion paragraph in the **Discussion** and focused on the convergent evolution instead of pathogenicity in that paragraph. This is related to comment #2 above. Again, we very much appreciate the critical suggestions for improving our manuscript.

Reviewer #3 (Remarks to the Author):

Ito and colleagues have provided a detailed investigation of convergent evolution in the spike protein of SARS-CoV-2 Omicron subvariants and their functional impacts on viral behaviors and pathogenesis. This is an intriguing study given the integration of modelling approaches to estimating viral fitness impacts for spike protein mutations. Importantly, this was integrated with functional investigations using both pseudotyped viruses and wild-type viruses. This is an impactful study for our understanding of viral evolution.

Our reply:

First of all, we are very happy to hear your positive feedback about our study and the impact it may have. The revised manuscript was modified according to reviewer's suggestions.

However, there are some key points of consideration for the authors that should be addressed. These are detailed below:

-Some of the grammar is confusing in sections, such as lines 260-262 with "either of these three substitutions". This becomes clear in a the following sentences (cooperative contributions vs individual activities); however, this could be more precisely worded up front for reader clarity.

Our reply:

We apologize for our insufficient English ability and resulting confusion. The revised manuscript has been proofread by multiple native English speakers. We hope the revised manuscript has improved clarity.

-in lines 269-270: A question here is how the sera from different breakthrough infection cases is being normalized or standardized. For example, in an instance where an individual has a below-average response to vaccination in regard to antibody generation, how would this sera compare to that from a strong or average responder?

Our reply:

In this experiment, we compared the NT_{50} values (serum dilution which provided 50% inhibition) of each individual serum to the pseudoviruses bearing respective S proteins (e.g., BQ.1.1 or BA.5). I.e., we compared the values "between pseudoviruses" but did not compare the values "between serum donors". Statistically significant differences were determined by a two-sided Wilcoxon signed-rank test, which is a non-parametric, paired test. This method is commonly used for SARS-CoV-2 research and our previous studies used the completely same method. Therefore, we feel this is a standard protocol to measure the antiviral activity of human sera.

-in lines 287-288: How well can hamster sera be used for validation here given that the neutralization activities of BA.2, BA.5 and BA.2.75 were largely completely negated for other subvariants. How is this reflective of human sera and does this have implications for the robustness of these types of data?

Our reply:

In the case of human sera, the immunological background is very complicated and different to each other considering natural infections of other pathogens including human coronaviruses in the past, SARS-CoV-2 vaccination, and natural infections of a variety of SARS-CoV-2 variants. Therefore, evaluating the neutralization activity induced by the infection of a specific variant of SARS-CoV-2 is difficult when we use human sera. On the other hand, sera obtained from laboratory animals infected with a single strain of a virus will be useful for the antigenic comparison among emerging variants, and such investigations have been performed in influenza virus research to illustrate the antigenic cartography (e.g., Smith et al., Science, 2004. PMID: 15218094). In our case, the hamster sera infected with each of the variants were firstly used to verify its neutralization activity against the same variant (e.g., hamster sera infected with BA.2 showed neutralization activity against BA.2). We have modified the text for clearer explanation. Specifically, we modified the sentence explaining this experiment (lines 301-304, page 9): *“the hamster sera infected with BA.2, BA.5 or BA.2.75 showed neutralization activity against the variant of virus infected”*. Also, to clearly show this, the order of pseudoviruses used for neutralization assay was modified: in **Fig. 2c** of revised manuscript, the variant of virus infected was put on the left-most column in each figure panel.

-in the ACE2 Binding Affinity section, the following consideration should be noted. The definition of infection relies on both entry and multiplication of a pathogen. Are these replication-competent pseudotypes? If not, this is an assessment of binding and entry, which may correlate with infectivity, but that needs to be addressed with replication kinetics.

Our reply:

The pseudoviruses used in our study are not replication competent. To address replication kinetics, as mentioned, a replication competent system is needed. For that, recombinant viruses would be needed. If we try to satisfy this concern, we would have to generate at least eight recombinant viruses, which were used in **Fig. 4a,b** of the revised manuscript (**Fig. 3a,b** of the original manuscript). It would be technically possible, but it is very time-consuming (it would take ~3-4 months to finish). Moreover, we think the evaluation of the impact of each substitution in the spike protein (e.g., R346T, K444T and N460K) on viral replication kinetics would be beyond the scope of this study and should be addressed in a future investigation.

-lines 356-361: It would be prudent for the authors to provide some comment on the differences in replication kinetic dynamics between the iPSC-derived airway ECs and the ALI NHBEs.

Our reply:

The differences in replication kinetic dynamics between the iPSC-derived airway epithelial cells and the AO-ALI are likely due to the differences in the characteristics of these cells. iPSC-derived airway epithelial cells are differentiated from iPS cells, while AO-ALI are differentiated from normal human bronchial epithelial cells. Therefore, it is expected that the maturity and cell population of iPSC-derived airway epithelial cells are different from those of AO-ALI. Differences in replication kinetic dynamics may be due to differences in maturity and cell population of these cells. In the revised manuscript, we explain it in the **Results** section (lines 440-446, page 14).

-lines 363-371: The discussion on barrier disruption should be verified by a secondary or alternative method. Viruses can move from epithelial to endothelial barriers without appreciable differences in cell-cell contacts as demonstrated by TEER or ECIS experiments. Disruption could be addressed through these methods, fluroscein-based experiments, or microscopy

Our reply:

We have already confirmed that the percentage of viruses that invaded the vascular channel of airway-on-a-chip correlates with the permeability of fluorescent dye and immunostaining analysis (<https://www.biorxiv.org/content/10.1101/2022.08.05.502758v1>). Please find these results in the figure below. These data were obtained in our paper which is going under review in another journal, but we would be happy to share this manuscript if necessary. In that paper, we used airway-on-a-chip and B.1.1, BA.1, BA.2 and BA.5, and evaluated the percentage of viruses that invaded the vascular channel (**Fig. A** below), the permeability of fluorescent dye (**Fig. B** below), and VE-cadherin expression (**Fig. C** below). In the revised manuscript, we explained it in the result section (lines 444-446, page 13).

Fig. The endothelial barrier functions of the SARS-CoV-2-infected airway-on-a-chip. The airway-on-a-chips were infected with B.1.1, BA.1, BA.2, and BA.5. (A) The copy numbers of viral RNA in the top and bottom channels of an airway-on-a-chip were quantified. The percentage of viral RNA load in the bottom channel per top channel is shown. (B) The medium containing Lucifer yellow was injected into the top channel of an airway-on-a-chip. The amount of Lucifer yellow in the bottom channel was measured, and then the apparent permeability (P_{app}) coefficient was calculated. (C) Immunostaining analysis of VE-cadherin (red) in an airway-on-a-chip was performed. Nuclei were counterstained with DAPI (blue).

-lines 374-375: There is no description provided here of what was inoculated, how, route, etc. There should be some description provided here for species and route.

Our reply:

Sorry for the insufficient explanation. In the revised manuscript, we have mentioned them (lines 458-460, page 14).

-in line 398, the authors mention "spreading efficacy". What is spreading efficacy and how is this assessed? Do the authors mean viral dissemination?

Our reply:

Yes. According to the suggestion, we have rephrased "spreading efficacy" with "viral dissemination" in the revised manuscript (line 483, page 15).

Reviewer #4 (Remarks to the Author):

In this study, Ito et al investigate BQ.1.1, a recently emerged variant of SARS-CoV-2 which has expanded to become the currently predominating strain globally (with BQ.1). In particular, this variant is notable for descending from, and displacing, BA.5, which was the previously globally dominant variant. Given this dominance, studies of this variant are warranted, in order to understand its virological characteristics as well as its evolutionary basis. While there have been a number of other studies that have been published on the former, the focus on evolution, which is the main topic of this manuscript, has not been reported to this reviewer's knowledge, and this study is therefore an important contribution to the field.

The authors identify that many BA.5 descendants harbor mutations within the same residues in the spike protein, including BQ.1.1, and they therefore examine these residues using a phylogenetic approach, finding that several sites confer a fitness advantage for the virus. Notably, these analyses suggest that the mutations that led to BQ.1.1 are K444T, N460K, and R346T in the spike, in this order. The authors then test the virological characteristics of BQ.1.1. They first examine the serum neutralization of BQ.1.1 and the aforementioned three point mutations in the background of BA.5, finding that BQ.1.1 is the most resistant and that its serum resistance is not attributable to a particular single mutation. They next examine the ACE2 affinity, finding that the heightened affinity of BQ.1.1 is attributable to N460K, and that this correlated with its increased infectivity and fusogenicity. BQ.1.1 displayed greater growth kinetics in some models, whereas in others it displayed no difference or had decreased growth. Finally, in an in vivo hamster model, BQ.1.1 was comparable to BA.5 in terms of pathogenicity and spreading efficiency.

This study is of high quality and well-written, and the data are presented in a clear manner. The Sato Lab and their consortia members have made many contributions to the SARS-CoV-2 field and this study continues their outstanding work. The manuscript is appropriate for this journal and should be accepted, but this reviewer has some suggestions that may improve the manuscript, outlined below.

Our reply:

First of all, we greatly appreciate you feel the manuscript is well written and should be accepted. Also, we are glad to hear you recognize our G2P-Japan consortium. The revised manuscript has been modified according to reviewer suggestions.

Major comments:

- While the focus of this study is understandably the spike protein, BQ.1.1 carries several additional mutations in other genes (as noted by the authors in Fig 1b). A thorough virological analysis of these mutations is certainly outside the scope of this work, but where do these mutations rank within the phylogenetic analyses that were conducted? That is, are these mutations partially involved in the evolutionary trajectory of and rise of BQ.1.1, or is it only the spike mutations that are important?

Our reply:

Although BQ.1.1 possesses six substitutions in the non-S region when compared to the majority of parental BA.5, these substitutions except for NSP13:N268S are not specific to BQ.1.1 and were acquired in the ancestral lineages of BQ.1.1, which did not show a particularly higher R_e compared with BA.5 (**Extended Data Fig. 1a; Fig. R3**). Furthermore, these non-S substitutions were not recurrently acquired during the Omicron evolution. Therefore, we did not focus on the non-S substitutions in the present study.

We specified that most of these non-S substitutions are not BQ.1.1-specific in the revised manuscript.

Fig. R3. Non-S substitutions detected in BQ.1.1 compared to the majority of BA.5.

Although five non-S substitutions were detected, the substitutions other than NSP13:N268S were acquired in the ancestral lineages of BQ.1.1 and are not specific to BQ.1.1.

- An important and unique takeaway of this study is that BQ.1.1 arose through K444T, N460K, and R346T in order, according to the phylogenetic analyses presented in Fig 1. This is of course of importance as it furthers our understanding of how SARS-CoV-2 is evolving in the population, and why it may be doing so in this particular manner. However, the subsequent experiments and data seem to shy away from this evolutionary perspective. This reviewer suggests that some experimental evidence for this proposed evolutionary route would greatly strengthen this manuscript and differentiate it from the others that have been published, as similar data to what are currently presented in Figs. 2-3 have been described by other groups. One suggestion would be to repeat some of the experiments in Fig 2 and in Fig 3 with the inclusion of the combinatorial mutants (e.g., K444T + N460K, K444T + R346T, N460K + R346T). This **should** not be

too much additional work, but would serve to validate the phylogenetic analysis. It may also shed light on the role of K444T, which, in the currently presented dataset, does not seem to have a specific functional role, yet is suggested to have arose first in the rise of BQ.1.1.

Our reply:

Thank you for the proposal of intriguing experiments. However, we have decided not to pursue them for the following reasons: While we showed that the BQ.1.1 lineage acquired K444T, N460K, and R346T substitutions in this order, we did not claim that this order is significant or commonly seen. This is because the sequential acquisition of multiple substitutions is so rare that we were unable to analyze it statistically in our dataset as shown in **Fig. 1c** in the revised manuscript. Therefore, it is difficult to determine whether the actual order of the acquired substitutions was of any significance in the evolutionary process, or BQ.1.1's phenotype was simply a result of it (eventually) having all three substitutions. In fact, our phylogenetic analysis provides no salient patterns of how substitution acquisition is ordered in other lineages, and our modeling analysis suggests that the substitutions most likely have an additive effect on R_e (regardless of order). Finally, the main focus of the experimental part of the present study is to uncover the virological properties of BQ.1.1 itself. Experimentally exploring the molecular mechanisms underlying evolutionary convergence seems outside the scope of the current manuscript, but certainly of great interest for future work.

• In a similar vein, are there any thoughts on how the emergence of BQ.1.1 mirrors what occurred previously with other SARS-CoV-2 variants, and whether there seems to be a particular type of route that the virus favors? That is, do certain mutations that confer certain phenotypes occur at a certain point in the virus' evolution? For example, the R346T mutation is reminiscent of the R346K mutation that arose in BA.1 to form BA.1.1, but is this coincidence or are we seeing a particular pattern?

Our reply:

Our modeling analyses suggest that each of the five convergent substitutions (including R346X) increased viral fitness (**Fig. 2b** of revised manuscript) in all Omicron sublineages, likely explaining their recurrence. Furthermore, previous studies including ours (e.g., PMID: 36272413; PMID: 36198317; PMID: 36535326) show that these substitutions associate with the increased ability to escape from humoral immunity (e.g., R346K/T and F486V) and/or to bind to ACE2 (e.g., L452R and N460K), suggesting that these substitutions increased viral fitness through increasing these abilities of S protein. Interestingly, the sites under convergent evolution might be virus context-dependent. For example, Martin et al. show that S substitution N501Y was being convergently acquired in earlier VOCs alpha, beta and gamma (PMID: 34537136). We now include this citation and point in our revised **Discussion** section which should focus more on

the importance of convergence in SARS-CoV-2's evolution (line 561–577, page 17).

Minor comments:

- Line 145: “principals” should be “principles”
- Line 193: Why were Omicron sequences only used from the UK and not globally?
- Line 371: “fusogenic” should be “fusogenicity”
- Line 1498: There seems to be an error with this reference
- Fig 1i: The x-axis has the label “2023/9” on the right panel; should this not be “2022”?

Our reply:

We have applied your suggested corrections.

Particularly, to answer the following concern:

- Line 193: Why were Omicron sequences only used from the UK and not globally?

The status of variant epidemics and genome surveillance (e.g., sampling rate) varies from country to country. Therefore, simply mixing data from different countries and estimating viral fitness based on the mixed data may lead to some bias in the estimated values. Therefore, in this study, we conducted the analysis using data from a single country (the UK).

In the revised manuscript, we added an analysis to assess the extent to which the model trained on the UK data can predict the fitness of variants from other countries (**Fig. 2d,e and Extended Data Fig. 2**).

Reviewers' Comments:

Reviewer #2:

Remarks to the Author:

Authors have addressed my major concerns.

In the abstract, "evades breakthrough BA.2/5 infection sera more efficiently" is not clear. Do you mean "BA.2 and BA.5" ? Please just write it that way.

Also, in the abstract, I wouldn't use "variant soups". Something like "governing the convergent evolution for currently known Omicron lineages" is enough.

Reviewer #3:

Remarks to the Author:

I'd like to thank the authors for their thoughtful answers and considerations of the points raised during the review. They have addressed these concerns in their revision and rebuttal document.

Reviewer #4:

Remarks to the Author:

The authors have addressed my previous comments and the manuscript is suitable for publication.

Reviewer #2 (Remarks to the Author):

Authors have addressed my major concerns.

In the abstract, "evades breakthrough BA.2/5 infection sera more efficiently" is not clear. Do you mean "BA.2 and BA.5" ? Please just write it that way.

Also, in the abstract, I wouldn't use "variant soups". Something like "governing the convergent evolution for currently known Omicron lineages" is enough.

Our reply: according to the reviewer's suggestion, we modified the sentences in the abstract.

Reviewer #3 (Remarks to the Author):

I'd like to thank the authors for their thoughtful answers and considerations of the points raised during the review. They have addressed these concerns in their revision and rebuttal document.

Our reply: we thank the reviewer for accepting our revision.

Reviewer #4 (Remarks to the Author):

The authors have addressed my previous comments and the manuscript is suitable for publication.

Our reply: we thank the reviewer for accepting our revision.